# TEMPO: Prompt-based Generative Pre-trained Transformer for Time Series Forecasting

**Defu Cao[1], Furong Jia[1], Sercan Ö. Arık[2], Tomas Pfister[2], Yixiang Zheng[1], Wen Ye[1], Yan Liu[1]**

[1] University of Southern California

[2] Google Cloud AI Research

{defucao, florajia, yixiangzheng, yewen, yanliu.cs}@usc.edu

{soarik, tpfister}@google.com,

## ABSTRACT

The past decade has witnessed significant advances in time series modeling with deep learning. While achieving state-of-the-art results, the best-performing architectures vary highly across applications and domains. Meanwhile, for natural language processing, the Generative Pre-trained Transformer (GPT) has demonstrated impressive performance via training one general-purpose model across various textual datasets. It is intriguing to explore whether GPT-type architectures can be effective for time series, capturing the intrinsic dynamic attributes and leading to significant accuracy improvements. In this paper, we propose a novel framework, TEMPO, that can effectively learn time series representations. We focus on utilizing two essential inductive biases of the time series task for pre-trained models: (i) decomposition of the complex interaction between trend, seasonal and residual components; and (ii) introducing the design of prompts to facilitate distribution adaptation in different types of time series. TEMPO expands the capability for dynamically modeling real-world temporal phenomena from data within diverse domains. Our experiments demonstrate the superior performance of TEMPO over state-of-the-art methods on zero shot setting for a number of time series benchmark datasets. This performance gain is observed not only in scenarios involving previously unseen datasets but also in scenarios with multi-modal inputs. This compelling finding highlights TEMPO's potential to constitute a foundational model-building framework.

## 1 INTRODUCTION

Time series forecasting, i.e., predicting future data based on historical observations, has broad real-world applications, such as health, transportation, finance and so on. In the past decade, numerous deep neural network architectures have been applied to time series modeling, including convolutional neural networks (CNN) (Bai et al., 2018), recurrent neural networks (RNN) (Siami-Namini et al., 2018), graph neural networks (GNN) (Li et al., 2018; Cao et al., 2021), and Transformers (Liu et al., 2021; Zhou et al., 2021; Wu et al., 2023; Zhou et al., 2022; Woo et al., 2022; Kitaev et al., 2020; Nie et al., 2023), leading to state-of-the-arts results. While achieving strong prediction performance, some of the previous works on time series mostly benefit from the advance in sequence modeling (from RNN and GNN, to transformers) that captures temporal dependencies but have not fully capitalized on the benefits of intricate patterns within time series data, such as seasonality, trend, and residual. These components are the key differentiating factors of time series from classical sequence data (Fildes et al., 1991). As a result, recent studies suggest that deep learning-based architectures might not be as robust as previously thought and might even be outperformed by shallow neural networks or even linear models on some benchmarks (Zeng et al., 2023; Zhang et al., 2022b; Wu et al., 2023; Ekambaram et al., 2023; Fan et al., 2022). Despite the notable success of deep learning forecasters, the vast majority of them still follow a conventional training mechanism, training and predicting using the same datasets.

Meanwhile, the rise of foundation models in natural language processing (NLP) and computer vision (CV), such as LLaMA (Touvron et al., 2023), CLIP (Radford et al., 2021) and ChatGPT, marks

major milestones on effective representation learning. It is extremely intriguing to explore a pre-trained path for foundation time series models with vast amounts of data, facilitating performance improvement in downstream tasks. Some recent works shed light into the possibility of building general transformers for time series (Zhou et al., 2023; Sun et al., 2023; Goswami et al., 2024; Das et al., 2023b; Rasul et al., 2023). However, the theoretical and practical understanding of such models has not reached the consensus observed in other domains where generative models have been widely acknowledged (Garza & Mergenthaler-Canseco, 2023). In addition, prompting techniques in LLM (such as InstructGPT (Ouyang et al., 2022)) provide a way to leverage the model's existing representations during pre-training instead of requiring learning from scratch. However, existing backbone structures and prompt techniques in language models do not fully capture the evolution of temporal patterns as in N-BEATS (Oreshkin et al., 2019) and AutoFormer (Wu et al., 2021), which are fundamental for time series modeling.

In this paper, we make an attempt to address the timely challenges of adapting large pre-trained models for time series forecasting tasks and developing a prompt-based generative pre-training transformer for time series, namely TEMPO. TEMPO consists of two key analytical components for effective time series representation learning: *one* focuses on modeling specific time series patterns, such as trends and seasonality, and *the other* concentrates on obtaining more universal and transferrable insights from the inherent properties of data through a prompt-based approach. Specifically, TEMPO firstly decomposes time series input into three additive components, i.e., trend, seasonality, and residuals via locally weighted scatterplot smoothing (Cleveland et al., 1990). Each of these temporal inputs is subsequently mapped to its corresponding hidden space to construct the time series input embedding of the generative pre-trained transformer (GPT). We conduct a formal analysis, bridging the time series domain with the frequency domain, to highlight the necessity of decomposing such components for time series analysis. In addition, we theoretically reveal that the attention mechanism is hard to achieve the decomposition automatically. Second, TEMPO utilizes a soft prompt to efficiently tune the GPT (Radford et al., 2019) for forecasting tasks by guiding the reuse of a collection of learnable continuous vector representations that encode temporal knowledge of trend and seasonality. In addition, we leverage the three key additive components of time series data—trend, seasonality, and residuals— to provide an interpretable framework for comprehending the interactions among input components (Hastie, 2017). Experiment results on zero shot setting and multimodal setting of TEMPO pave the path to foundational models for time series. Besides, we demonstrate the stable predictive power of our model on unseen samples with textual information on two multimodal datasets including TETS (Text for Time Series) dataset, which is first introduced in this work to foster further research topics of pre-trained time series models.

In summary, the main **contributions** of our paper include: (1) We introduce an interpretable prompt-tuning-based generative transformer, TEMPO, for time series representation learning. It further drives a paradigm shift in time series forecasting - from conventional deep learning methods to pre-trained foundational models. (2) We adapt pre-trained models for time series by focusing on two fundamental inductive biases: First, we utilize decomposed trend, seasonality, and residual information. Second, we explore the soft prompt strategies to accommodate time series data's dynamic nature. (3) Through extensive experimentation on benchmark datasets and two multimodal datasets, our model demonstrates superior performance. Notably, our robust results towards highlights the potential of foundational models in the realm of time series forecasting.

## 2 RELATED WORKS

**Pre-trained Large Language Models for Time Series.** The recent development of Large Language Models (LLMs) has opened up new possibilities for time-series modeling. LLMs, such as T5 (Raffel et al., 2020), GPT (Radford et al., 2018), GPT-2 (Radford et al., 2019), GPT-3 (Brown et al., 2020), GPT-4 (OpenAI, 2023), LLaMA (Touvron et al., 2023), have demonstrated a strong ability to understand complex dependencies of heterogeneous textual data and provide reasonable generations. Recently, there is growing interest in applying language models to time series tasks (Jin et al., 2024a; Gruver et al., 2024). For example, Xue & Salim naively convert time series data to text sequence inputs and achieves encouraging results. Sun et al. propose text prototype-aligned embedding to enable LLMs to handle time series data. In addition, Yu et al. present an innovative approach towards leveraging LLMs for explainable financial time series forecasting. The works in (Zhou et al., 2023) and (Chang et al., 2023) are the most relevant ones to our work, as they both introduce approaches for

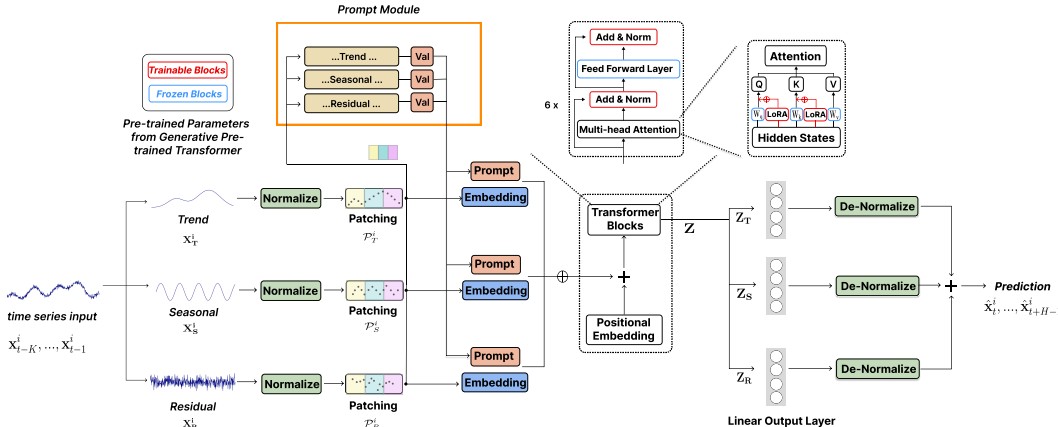

Figure 1: The architecture of proposed TEMPO-GPT. The trend $X_T$, seasonal $X_S$ and residual $X_R$ components are treated as different semantic inductive biases to feed into the pre-trained transformer.

time-series analysis by strategically leveraging and fine-tuning LLMs. However, these studies directly employ time series data to construct embeddings, without adequately capturing the inherent and unqiue characteristics of time series data which is challenging to decouple such information within the LLMs (Shin et al., 2020). In addition, there is still very limited work on LLM for multimodal data with time series. METS (Li et al., 2023) is one of the early works pursuing this direction. While the experiment results are encouraging, it is difficult to extend METS to other modalities since the embedding alignment between time series and texts are specific. Please refer to the suvery papers (Jin et al., 2023; 2024b) for further references of time series meeting LLMs.

**Prompt tuning.** Prompt tuning is an efficient, low-cost way of adapting a pre-trained foundation model to new downstream tasks which has been adapted to downstream tasks across various domains. In NLP domain, soft prompts with trainable representation are used through prompt-tuning (Lester et al., 2021) or prefix-tuning (Li & Liang, 2021). Prompting techniques have also been extended to CV tasks like object detection(Li et al., 2022) and image captioning (Zhang et al., 2022a), etc and other domains such as misinformation (Zhang et al., 2024). Multimodal works, such as CLIP (Radford et al., 2021), use textual prompts to perform image classification and achieve SOTA performance. In addition, L2P (Wang et al., 2022b) demonstrates the potential of learnable prompts stored in a shared pool to enable continual learning without rehearsal buffer, and Dualprompt (Wang et al., 2022a) introduces a dual-space prompt architecture, maintaining separate prompt encodings for general knowledge and expert information, etc. Our research builds upon these concepts by exploring the use of prompt design from indicative bias specifically for temporal reasoning and knowledge sharing across time series forecasting problems.

## 3 METHODOLOGY

In our work, we adopt a hybrid approach that incorporates the robustness of statistical time series analysis with the adaptability of data-driven methods. As shown in Figure 1, we propose a novel integration of seasonal and trend decomposition from STL (Cleveland et al., 1990) into the pre-trained transformers. This strategy allows us to exploit the unique strengths of both statistical and machine learning methods, enhancing our model's capacity to handle time series data efficiently. Moreover, a semi-soft prompting approach is introduced to enhance the adaptability of pre-trained models for handling time series data. This innovative approach enables the models to merge their extensive learned knowledge with the unique requirements intrinsic to time series analysis.

### 3.1 PROBLEM DEFINITION

Given observed values of previous $K$ timestamps, the task of multivariate time-series forecasting aims to predict the values for the next $H$ timestamps. That is,

$$\hat{\mathbf{x}}_t^i, ..., \hat{\mathbf{x}}_{t+H-1}^i = F(\mathbf{x}_{t-K}^i, ..., \mathbf{x}_{t-1}^i; \mathbf{V}^i; \Phi) \tag{1}$$

where $\hat{\mathbf{x}}_t^i, ..., \hat{\mathbf{x}}_{t+H-1}^i$ is the vector of $H$-step estimation from timestamp $t$ of channel $i$ corresponding to the $i$-th feature. Given the historical values $\mathbf{x}_{t-K}^i, ..., \mathbf{x}_{t-1}^i$, it can be inferred by model $F$ with parameter $\Phi$ and prompt $\mathbf{V}^i$. In anticipation of the foundational model's strong generalization capabilities across unseen datasets, we default to a zero-shot learning configuration in the absence of specific indications. This approach entails that the model is not privy to the target dataset's history value and horizon value during the training process.

## 3.2 TIME SERIES INPUT REPRESENTATION

For time series data, representing the complex input by decomposing it into meaningful components, such as trend and season components, can help extract information optimally. In this paper, given the input $X \in \mathbb{R}^{n \times L}$, where $n$ is the feature (channel) size and $L$ it the length of the time series, the additive STL decomposition (Cleveland et al., 1990) can be represented as:

$$X^i = X_T^i + X_S^i + X_R^i. \tag{2}$$

Here, $i$ is the channel index (corresponding to a certain covariate) for multivariate time series input, and the trend $X_T \in \mathbb{R}^{n \times L} = \frac{1}{m} \sum_{j=-k}^k X_{t+j}$ captures the underlying long-term pattern in the data, where $m = 2k + 1$ and $k$ is the averaging step size. The seasonal component $X_S \in \mathbb{R}^{n \times L}$ encapsulates the repeating short-term cycles, which can be estimated after removing the trend component. The residual component $X_R \in \mathbb{R}^{n \times L}$ represents the remainder of the data after the trend and seasonality have been extracted. Note that, in practice, it is suggested to leverage as much information as possible to achieve a more precise decomposition. However, in consideration of computational efficiency, we opt not to use the STL decomposition on the largest possible data window on each instance. Instead, we perform local decomposition within each instance using a fixed window size. Inspired by N-BEATs (Oreshkin et al., 2019), we introduce learnable parameters for estimating the various local decomposition components. Same for the others. This principle applies to other components of the model as well. In Appendix G, we establish a connection between time series forecasting and frequency domain prediction, where our findings indicate that decomposition significantly simplifies the prediction process. Note that such decomposition is of more importance in current transformer-based methods as the attention mechanism, in theory, may not disentangle the disorthogonal trend and season signals automatically:

**Theorem 3.1** *Suppose that we have time series signal $X = X_{Tt} + X_{St} + X_{Rt}, t \in [t_1, t_n]$. Let $E = \{e_1, e_2, ..., e_n\}$ denote a set of orthogonal bases. Let $E_S \subseteq E$ denote the subset of $E$ on which $X_{St}$ has non-zero eigenvalues and $E_T \subseteq E$ denote the subset of $E$ on which $X_{Tt}$ has non-zero eigenvalues. If $X_{St}$ and $X_{Tt}$ are not orthogonal, i.e. $\sum_{i=1}^n X_{Tt}^i X_{St}^i \neq 0$, then $E_T \cap E_S \neq \emptyset$, i.e. $E$ can not disentangle the two signals onto two disjoint sets of bases.*

The proof can be found in Appendix G. Theorem 3.1 states that if trend and seasonal components of a time series are non-orthogonal, they cannot be fully disentangled and separated by any set of orthogonal bases. According to (Zhou et al., 2023), the self-attention layer naturally learns an orthogonal transformation, akin to PCA's decomposition into orthogonal principal components. Thus, applying attention directly to a raw time series would be ineffective at disentangling non-orthogonal trend and seasonal components. For the remainder of the methodology section, we will utilize the trend component $X_T$ as the exemplary case. We first apply reverse instance normalization (Kim et al., 2022) on each global component and local input respectively to facilitate knowledge transfer and minimize losses introduced by distribution shifts. That is, for each sample $x_{Tt}$ from $X_T$'s -th channel of time $t$, $\hat{x}_{Tt} = \gamma_T \left( x_{Tt} - \mathbb{E}_t \left[ x_{Tt} \right] / \sqrt{\text{Var} \left[ x_{Tt} \right] + \epsilon_T} \right) + \beta_T$, where $\mathbb{E}_t \left[ x_{Tt} \right]$ and $\text{Var} \left[ x_{Tt}^i \right]$ are the instance-specific mean and standard deviation; $\gamma_T$ and $\beta_T$ are trainable affine parameter vectors for trend component. In addition, we implement a mean square error (MSE) reconstruction loss function to ensure that the local decomposition aligns with the global STL decomposition observed in the training data. The decomposition loss function, denoted as $L_{Dec} = f_T(X, \theta_T) - \hat{X}_T^g$, where $f_T$ is the function with learnable variables $\theta_T$ for mapping local decomposition to be aligned with the global decomposition after normalization $\hat{X}_T^g$. Then, following (Nie et al., 2023), we combine time-series patching with temporal encoding to extract local semantics by aggregating adjacent time steps into tokens, significantly increasing the historical horizon while reducing redundancy. Specifically, we get the patched token for the $i$-th normalized trend component for $f_T(X^i, \theta_T)$ with

$P_T^i \in \mathbb{R}^{L_P \times N}$, where $L_P$ is the patch length, $N = \left\lfloor \frac{(L-L_P)}{S} \right\rfloor + 2$ is the number of patches and $S$ is the stride. We get patched tokens $P_S^i$ and $P_R^i$ in the same way. Then, we feed the patched time series tokens to the embedding layer $f$ to get the representation $\mathcal{P}_T^i = f(P_T^i) \in \mathbb{R}^{P \times L_E}$ for the language model architecture to transfer its language capabilities to the novel sequential modality effectively, where $L_E$ is the embedding size.

## 3.3 Prompt Design

Prompting techniques have demonstrated remarkable effectiveness across a wide range of applications by leveraging the power of task-specific knowledge encoded within carefully crafted prompts. This success can be attributed to the prompts' ability to provide a structured framework that aligns the model's outputs with the desired objectives, resulting in enhanced accuracy, coherence, and overall quality of the generated content. Previous works mostly focus on utilizing a fixed prompt to boost the pre-trained models' performance through fine-tuning (Brown et al., 2020). In pursuit of leveraging the rich semantic information encapsulated within various time series components, our research introduces a semi-soft prompting strategy. This approach involves the generation of distinct prompts corresponding to each primary time series component: trend, seasonality, and residuals. *'Predict the future time step given the [trend, season, residual]'* serves as the template from which we derive our component-specific prompts. These are subsequently concatenated with the relevant component data, thereby enabling a more refined modeling approach that acknowledges the multifaceted nature of time series data. Specifically, commence by translating the trend-specific prompts into the word embedding space, followed by a linear transformation to derive the learnable trend prompt vector $V_t$. This so-called 'semi-soft' prompt design thus strikes a balance between the interpretability and initial guidance of a 'hard' prompt and the adaptability of a 'soft' prompt. The combined embedding of this prompt with the time series representation is encapsulated by:

$$\boldsymbol{x}_T = [V_t; \mathcal{P}_T] \tag{3}$$

Here, $\boldsymbol{x}_T$ denotes the aggregation of embeddings along the temporal axis. This concatenation procedure is mirrored for the seasonality and residual components, yielding $\boldsymbol{x}_S$ and $\boldsymbol{x}_R$, respectively. This framework allows for an instance to be associated with specific prompts as the inductive bias, jointly encoding critical information relevant to the forecasting task, such as recurring patterns, overarching trends, and inherent seasonality effects. It is of note that our prompt design maintains a high degree of adaptability, ensuring compatibility with a broad spectrum of time series analyses. In particular, similar with (Wang et al., 2022a), we introduce prompt pool as an extension of our design of soft prompt in Appendix D, aimed at accommodating the characteristically non-stationary nature of real-world time series data and the associated distributional shifts (Huang et al., 2020; Fan et al., 2023). This adaptability underscores the potential of our prompting strategy to evolve in congruence with the complexities presented by diverse time series datasets.

## 3.4 Generative Pre-trained Transformer Architecture

We use the decoder-based generative pre-trained transformer (GPT) as the backbone to build the basis for the time-series representations. To utilize the decomposed semantic information in a data-efficient way, we choose to concatenate the prompt and different components together and put them into the GPT block. Specifically, the input of our time series embedding can be formulated as: $\boldsymbol{x} = \boldsymbol{x}_T \oplus \boldsymbol{x}_S \oplus \boldsymbol{x}_R$, where $\oplus$ corresponds to concatenate operation and $\boldsymbol{x}_*$ can be treated as different sentences. Note that, another alternative way is to build separate GPT blocks to handle different types of time series components. Inside the GPT block, we adopt the strategy used in (Zhou et al., 2023) and opt to update the gradients of the position embedding layer and layer normalization layers. In addition, we employ LORA (Low-Rank Adaptation) (Hu et al., 2021) to adapt to varying time series distributions efficiently as it performs adaptation with significantly fewer parameters.

The overall forecasting result should be an additive combination of the individual component predictions. Finally, the outputs $Z$ of $n$ features from the GPT block can be split into $Z_T, Z_S, Z_R \in \mathbb{R}^{n \times P \times L_E}$ (output corresponding to trend, seasonality, and residual) based on their positions in the input order. Each $Z$ component is then fed into fully connected layers to generate predictions $Y_* \in \mathbb{R}^{n \times L_H}$, where $L_H$ is the prediction length. The forecast results can be formulated as follows: $\hat{Y} = \hat{Y_T} + \hat{Y_S} + \hat{Y_R}$. After that, we de-normalize $Y$ according to the corresponding statistics used in

the normalization step: $\hat{Y}_t^i = \sqrt{\text{Var}\left[x_t^i\right] + \epsilon} \cdot \left(\frac{Y_t^i - \beta}{\gamma}\right) + \mathbb{E}_t\left[x_t^i\right]$. By recombining these additive elements, our approach aims to reconstruct the full temporal trajectory most representative of the underlying dynamics across varied timescales captured by the decomposed input representation.

In order to achieve interpretability, we explore both linear and nonlinear interactions among trend, seasonal, and residual components in their contribution to the final output. Therefore we construct an interpretable generalized additive model (GAM) (Hastie, 2017) based on GPT's output to learn how the three components interact with each other, which is: $g(Y) = F_\emptyset + \sum_i F_i(x_i) + \sum_t F_{\mathcal{I}_t}(x_{\mathcal{I}_t})$, where $F_\emptyset$ is a normalizing constant, the footnote $i$ corresponds to the trend, season, and residual component. $\{\mathcal{I}_t\}$ is of a set of multiple interact components. Then, we can calculate the first-order sensitivity index (Sobol', 1990) or SHAP (SHapley Additive exPlanations) value (Lundberg & Lee, 2017) to measure the sensitivity of each component.

## 4 EXPERIMENTS

Our experiments are conducted using widely-recognized time series benchmark datasets, such as those detailed in (Zhou et al., 2021), alongside the GDELT dataset (Jia et al., 2024) and our proposed TETS dataset. These comprehensive datasets encompass a diverse array of domains, including, but not limited to, electricity (ETTh1, ETTh2, ETTm1, ETTm2, Electricity), traffic (Traffic), climate (Weather), news (GDELT), and finance (TETS), with data sampling frequencies ranging from minutes, hours to days and quarters. The inclusion of such varied datasets ensures a thorough evaluation of our experimental setups across multiple dimensions of time series data. Due to the absence of a standard test split for zero-shot comparison, we adopt a uniform training methodology to ensure fair performance assessment across datasets unseen during model training. Specifically, to advance the paradigm of foundation models within the domain of transfer learning, we investigate a zero-shot setting for our experiments, which is the 'many-to-one' scenario: training on multiple source datasets followed by zero-shot forecasting on a distinct, unseen target dataset. For instance, when evaluating performance on a 'weather' dataset, our model is pre-trained on diverse datasets including 'ETTm1, ETTm2, ETTh1, ETTh2, Electricity, and Traffic' without exposure to the target weather data. This 'many-to-one' approach differs fundamentally from 'one-to-one' or 'one-to-many' configurations (Zhang et al., 2022c) by using diverse pre-training datasets from varied domains, like traffic and weather data. This diversity, while rich, introduces complexity, as the model must identify patterns across potentially misaligned samples, complicating learning compared to models trained and tested on in distribution datasets.

We use GPT-2 (Radford et al., 2019) as our backbone to build TEMPO[1] as shown in Figure 1. To comprehensively demonstrate the performance of our model, we compare TEMPO with the following baselines over long-term forecasting and short-term forecasting: (1) The pre-trained LLM-based models, including Bert (Devlin et al., 2019), GPT2 (Radford et al., 2019; Zhou et al., 2023), T5 (Raffel et al., 2020), and LLaMA (Touvron et al., 2023). (2) The Transformer-based models, including the PatchTST (Nie et al., 2023), FEDformer (Zhou et al., 2022), ETSformer (Woo et al., 2022) and Informer (Zhou et al., 2021). (3) The variant of Linear-based models, DLinear (Zeng et al., 2023) model. (4) General 2D-variation model, TimesNet (Wu et al., 2023). Following traditional forecasting works, we report the Mean Squared Error(MSE) and Mean Absolute Error (MAE) results in this section. Please refer to the Appendix B and F for the detailed experiment setting and baselines.

### 4.1 ZERO SHOT LONG-TERM FORECASTING RESULTS

Table 1 presents the performance of multiple time series forecasting models on MSE and MAE metrics across different prediction lengths under the 'many-to-one' setting, with lower scores indicating more accurate forecasts. Our proposed model, TEMPO, surpassed existing baselines on average over all prediction horizons across all datasets, highlighting the broad applicability of TEMPO. Our model achieves the highest average performance scores. Specifically, it improves the weather and ETTm1 datasets by around 6.5% and 19.1%, respectively in MAE compared to the previous state-of-the-art model, PatchTST. It also secures the lowest error rates across numerous individual dataset-prediction length configurations. Compared to other pre-trained models for forecasting, TEMPO consistently delivers the best results across different time series datasets. These results suggest that incorporating

---

[1]TEMPO's source code can be found at: https://github.com/DC-research/TEMPO

Table 1: Transfer learning of long-term forecasting results on time series benchmark datasets. We use prediction length $O \in \{96, 192, 336, 720\}$. A lower MSE indicates better performance. Hereafter, for the tables, the best results are marked in bold and the second optimal in underlined, respectively with MSE/MAE.

| Horizon | Model | ECL MSE/MAE | Traffic MSE/MAE | Weather MSE/MAE | Ettm1 MSE/MAE | Ettm2 MSE/MAE | Etth1 MSE/MAE | Etth2 MSE/MAE |
|---|---|---|---|---|---|---|---|---|
| 96 | TEMPO | **0.178/0.276** | **0.476/0.343** | **0.211/0.254** | **0.438/0.424** | **0.185/0.267** | **0.400/0.406** | **0.301/0.353** |
| | GPT2 | 0.193/0.288 | 0.522/0.380 | 0.226/0.274 | 0.486/0.438 | 0.193/0.273 | 0.400/0.416 | 0.320/0.363 |
| | T5 | 0.185/0.282 | 0.508/0.366 | 0.217/0.271 | 0.529/0.464 | 0.190/0.268 | **0.400**/0.409 | 0.328/0.366 |
| | PatchTST | 0.489/0.546 | 1.023/0.641 | 0.247/0.301 | 0.733/0.554 | 0.273/0.345 | 0.57/0.518 | 0.379/0.412 |
| | Timesnet | 0.293/0.369 | 0.585/0.401 | 0.247/0.295 | 0.518/0.470 | 0.202/0.290 | 0.407/0.423 | 0.315/0.362 |
| | FEDformer | 0.300/0.399 | 0.835/0.564 | 0.292/0.346 | 0.698/0.553 | 0.665/0.634 | 0.509/0.502 | 0.385/0.426 |
| | ETSformer | 0.707/0.638 | 1.419/0.795 | 0.453/0.416 | 1.117/0.678 | 0.353/0.404 | 0.469/0.457 | 0.405/0.428 |
| | Informer | 0.512/0.531 | 1.400/0.830 | 0.837/0.711 | 0.880/0.657 | 0.263/0.360 | 0.642/0.562 | 0.704/0.651 |
| | DLinear | 0.195/0.292 | 0.609/0.424 | 0.212/0.275 | 0.624/0.522 | 0.264/0.352 | 0.414/0.421 | 0.334/0.389 |
| 192 | TEMPO | **0.198/0.293** | **0.496/0.355** | **0.254/0.298** | **0.461/0.432** | **0.243/0.304** | **0.426/0.421** | **0.355/0.389** |
| | GPT2 | 0.207/0.300 | 0.533/0.387 | 0.273/0.312 | 0.516/0.461 | 0.254/0.312 | 0.441/0.433 | 0.381/0.402 |
| | T5 | 0.205/0.302 | 0.524/0.374 | 0.277/0.321 | 0.523/0.454 | 0.246/0.306 | 0.428/0.426 | 0.413/0.410 |
| | PatchTST | 0.465/0.535 | 0.992/0.633 | 0.277/0.324 | 0.739/0.563 | 0.299/0.355 | 0.580/0.528 | 0.387/0.417 |
| | Timesnet | 0.283/0.366 | 0.64/0.431 | 0.316/0.342 | 0.55/0.490 | 0.261/0.318 | 0.439/0.439 | 0.394/0.406 |
| | FEDformer | 0.390/0.468 | 0.869/0.579 | 0.372/0.426 | 0.819/0.608 | 0.358/0.416 | 0.683/0.596 | 0.921/0.748 |
| | ETSformer | 0.721/0.645 | 0.995/0.658 | 0.545/0.466 | 1.598/0.803 | 0.390/0.416 | 0.548/0.503 | 0.476/0.468 |
| | Informer | 0.625/0.619 | 0.872/0.506 | 0.431/0.455 | 1.461/0.892 | 0.494/0.516 | 0.798/0.632 | 0.455/0.883 |
| | DLinear | 0.204/0.300 | 0.595/0.412 | 0.259/0.308 | 0.599/0.511 | 0.292/0.365 | 0.439/0.437 | 0.381/0.415 |
| 336 | TEMPO | **0.209/0.309** | **0.503/0.356** | **0.292**/0.332 | **0.515/0.467** | **0.309/0.345** | **0.441/0.430** | **0.379/0.408** |
| | GPT2 | 0.231/0.324 | 0.566/0.421 | 0.441/0.379 | 0.571/0.502 | 0.315/0.35 | 0.449/0.440 | 0.394/0.416 |
| | T5 | 0.229/0.321 | 0.550/0.391 | 0.330/**0.330** | 0.572/0.504 | 0.316/0.346 | 0.442/0.438 | 0.416/0.427 |
| | PatchTST | 0.531/0.569 | 0.987/0.626 | 0.317/0.347 | 0.755/0.576 | 0.342/0.382 | 0.677/0.573 | 0.386/0.425 |
| | Timesnet | 0.733/0.633 | 1.609/0.864 | 0.359/0.372 | 0.638/0.532 | 0.38/0.392 | 0.555/0.503 | 0.384/0.413 |
| | FEDformer | 0.317/0.406 | 1.006/0.640 | 0.639/0.600 | 0.785/0.624 | 0.372/0.424 | 0.582/0.542 | -/5.755 |
| | ETSformer | 0.862/0.707 | 0.940/0.621 | 0.487/0.444 | 1.154/0.682 | 0.409/0.428 | 0.728/0.585 | 0.446/0.451 |
| | Informer | 1.222/0.863 | 0.978/0.507 | 0.370/0.412 | 0.949/0.631 | 0.788/0.622 | 1.125/0.810 | 1.389/0.848 |
| | DLinear | 0.231/0.325 | 0.624/0.427 | 0.304/0.342 | 0.622/0.534 | 0.361/0.411 | 0.463/0.464 | 0.471/0.482 |
| 720 | TEMPO | 0.279/0.355 | **0.538/0.376** | **0.370/0.379** | **0.591/0.509** | **0.386/0.395** | 0.443/**0.451** | 0.409/0.440 |
| | GPT2 | 0.262/**0.347** | 0.596/0.399 | 0.484/0.422 | 0.646/0.54 | 0.394/0.397 | 0.445/0.454 | 0.434/0.448 |
| | T5 | 0.266/0.351 | 0.578/0.404 | 0.528/0.451 | 0.694/0.568 | 0.394/0.397 | 0.443/0.458 | 0.425/0.440 |
| | PatchTST | 0.475/0.532 | 1.152/0.706 | 0.375/0.388 | 0.739/0.57 | 0.421/0.421 | 0.540/0.521 | 0.425/0.448 |
| | Timesnet | 1.166/0.859 | 1.974/0.971 | 0.423/0.405 | 0.723/0.577 | 0.399/0.409 | **0.438/0.461** | **0.394/0.431** |
| | FEDformer | 0.423/0.48 | 0.965/0.652 | 0.409/0.425 | 0.816/0.614 | 0.455/0.462 | 0.688/0.618 | 0.427/0.452 |
| | ETSformer | 0.666/0.640 | 0.798/0.518 | 0.592/0.506 | 1.038/0.665 | 0.444/0.438 | 0.615/0.561 | 0.446/0.466 |
| | Informer | 0.881/0.778 | 1.532/0.800 | 1.133/0.842 | 0.779/0.616 | 1.075/0.725 | 0.836/0.687 | 1.330/0.866 |
| | DLinear | **0.259**/0.352 | 0.623/0.42 | 0.363/0.389 | 0.639/0.559 | 0.515/0.490 | 0.467/0.481 | 0.639/0.559 |
| Avg | TEMPO | **0.216/0.308** | **0.503/0.358** | **0.282/0.316** | **0.501/0.458** | **0.280/0.328** | **0.428/0.427** | **0.361/0.398** |
| | GPT2 | 0.223/0.315 | 0.554/0.397 | 0.356/0.347 | 0.555/0.485 | 0.289/0.333 | 0.436/0.436 | 0.382/0.407 |
| | T5 | 0.221/0.314 | 0.540/0.384 | 0.338/0.343 | 0.58/0.498 | 0.287/0.329 | **0.428**/0.433 | 0.396/0.411 |
| | PatchTST | 0.49/0.545 | 1.039/0.652 | 0.304/0.340 | 0.741/0.566 | 0.334/0.376 | 0.592/0.535 | 0.394/0.425 |
| | Timesnet | 0.619/0.557 | 1.202/0.667 | 0.336/0.354 | 0.607/0.517 | 0.311/0.352 | 0.460/0.457 | 0.372/0.403 |
| | FEDformer | 0.358/0.439 | 0.919/0.609 | 0.428/0.449 | 0.780/0.600 | 0.463/0.484 | 0.616/0.565 | -/1.845 |
| | ETSformer | 0.750/0.664 | 1.038/0.648 | 0.519/0.458 | 1.227/0.707 | 0.399/0.422 | 0.590/0.527 | 0.443/0.453 |
| | Informer | 0.810/0.698 | 1.196/0.661 | 0.693/0.605 | 1.017/0.699 | 0.655/0.556 | 0.850/0.673 | 0.970/0.812 |
| | DLinear | 0.222/0.317 | 0.613/0.421 | 0.284/0.329 | 0.621/0.531 | 0.358/0.405 | 0.446/0.451 | 0.456/0.461 |

LLM with the well-designed prompt and implementing time series decomposition can contribute significantly to enhancing the accuracy and stability of zero-shot time series forecasting.

## 4.2 SHORT-TERM FORECASTING WITH CONTEXTUAL INFORMATION

**Dataset and metrics.** In this section, we introduce TETS, a new benchmark dataset built upon S&P 500 dataset combining contextual information and time series, to the community. Following (Cao et al., 2023), we choose the symmetric mean absolute percentage error (SMAPE) as our metric in this section. Moreover, the GDELT is also used to verify the effectiveness the our proposed method. Please refer to Appendix B.2 and Appendix B.3 for the detailed dataset setting of TETS and GDELT;

Table 2: SMAPE results of EBITDA from TETS and GDELT. The result of EBITDA includes outliers removed where SMAPE exceeds 0.8/0.9. The best results are marked in bold and the second optimal in underlined respectively with 0.8 & 0.9. (*Sectors*: CC: Consumer Cyclical; CD: Consumer Defensive; Ind: Industrials; RE: Real Estate; *Events*: 11: Disapprove; 17: Coerce; 19:Fight.)

| EBITDA Dataset | | | | | | | | | |
|---|---|---|---|---|---|---|---|---|---|
| Sectors | TEMPO | LLaMA | GPT2 | Bert | T5 | Informer | PatchTST | Reformer | DLinear |
| CC | **32.27/33.48** | 33.13/34.31 | 33.77/35.37 | 33.42/35.33 | 32.65/33.83 | 41.12/43.17 | 41.44/43.18 | 37.23/39.09 | 33.53/35.65 |
| CD | **25.9/26.25** | 26.34/26.62 | 26.86/27.15 | 27.34/28.3 | 26.44/26.79 | 35.65/36.08 | 31.6/31.98 | 29.93/30.36 | 27.01/28.04 |
| Ind | **26.7/27.42** | 27.17/27.98 | 27.9/28.63 | 27.89/28.95 | 27.3/28.12 | 34.83/35.87 | 33.84/34.87 | 30.23/31.28 | 27.59/28.84 |
| RE | **29.46/30.11** | 29.63/30.48 | 30.62/31.21 | 30.62/31.66 | 30.1/30.64 | 36.4/37.22 | 37.63/38.31 | 31.23/31.69 | 29.95/30.92 |
| GDELT Dataset | | | | | | | | | |
| 11 | **38.77** | 40.23 | 39.03 | 38.89 | 39.04 | 42.00 | 40.45 | 46.72 | 40.14 |
| 17 | **41.02** | 42.50 | 41.20 | 41.10 | 41.24 | 44.44 | 42.72 | 48.08 | 42.45 |
| 19 | **44.03** | 45.49 | 44.17 | 44.09 | 44.29 | 47.45 | 45.49 | 48.30 | 45.40 |

Appendix H for the proposed pipeline of collecting TETS dataset with both time series and textual information.

**Contextual Information.** In order to incorporate the contextual information into our proposed TEMPO, we leverage the built-in tokenization capabilities of the generative pre-trained transformer to derive embeddings of input text. Then, we utilize these text embeddings corresponding to each time series instance, $Text$, to construct soft prompts with learnable parameters and concatenate them at the beginning of the input embedding, that is, $x = Text \oplus x_T \oplus x_S \oplus x_R$. Where the $x_*$ for EBITDA is conducted with semi-soft prompt. This method is not strictly confined to our proposed model but can be feasibly applied in similar works to enhance their capability of handling and benefiting from contextual information. Comparisons with other design strategies of contextual information are provided in the Appendix D.4 for further reference.

**Results.** From the transfer learning perspective, we choose to report the setting of 'many-to-many', which means we train a model using in-domain sectors data and directly do the zero-shot test on all cross-domain sectors. The SMAPE results of using different baseline models and our model on the TETS dataset and GDELT dataset are listed in Table 2 which is also zero-shot setting as data samples from those sectors are not seen during the training stage. Examining the results across all sectors, our proposed model, which combines time series data with supplementary summary (contextual) data, outperforms all the baseline methods in cross-domain sectors. Besides, we observe that transformer-based architectures training from scratch, specifically tailored for time series analysis—such as PatchTST, Informer, and Reformer (Kitaev et al., 2020)—tend to underperform in comparison to transformers pre-trained on linguistic datasets. This performance discrepancy indicates that the parameter initialization derived from pre-trained language models confers a superior starting point for model optimization. Consequently, these pre-trained models exhibit enhanced capabilities and adaptability within zero-shot learning contexts. Furthermore, in instances where the time series data exhibits a strong correlation to other modalities, such as textual information, devising an effective strategy to amalgamate these distinct modalities could lead to enhanced performance gains.

## 5 ANALYSIS

### 5.1 ABLATION STUDY

The provided ablation study, Table 3, offers critical insights into the impact of the prompt and decomposition components on the performance of our model. In this table, the MSE and MAE on various datasets are reported for four scenarios: the original model configuration ('TEMPO'); the model without the prompt design and without decomposition, which is the setting of 'w/o Dec'; the model without prompt design ('w/o Pro') and the model without the decomposition loss alignment ('w/o Dec Loss'). Averagely, the exclusion of the prompt component leads to a deterioration in the model's predictive accuracy, indicating the prompt can be an important factor in enhancing the model's overall performance. The omission of decomposition loss typically results in a decline in model performance. Decomposition loss facilitates the use of a richer historical dataset, which

| | | TEMPO | w/o Dec | w/o Pro | w/o Dec Loss |
|---|---|---|---|---|---|
| | | MSE/MAE | MSE/MAE | MSE/MAE | MSE/MAE |
| ECL | 96 | **0.178/0.276** | 0.195/0.294 | 0.185/0.281 | 0.191/0.293 |
| | 192 | 0.198/**0.293** | 0.210/0.301 | **0.196**/0.295 | 0.205/0.305 |
| | 336 | **0.209/0.309** | 0.237/0.328 | 0.225/0.318 | 0.243/0.337 |
| | 720 | **0.279**/0.355 | 0.271/**0.351** | 0.269/0.359 | 0.262/0.353 |
| | Avg | **0.216/0.308** | 0.228/0.319 | 0.219/0.313 | 0.225/0.322 |
| Ettm1 | 96 | 0.438/**0.424** | 0.516/0.447 | 0.452/0.431 | **0.428**/0.425 |
| | 192 | **0.461/0.432** | 0.518/0.462 | 0.47/0.45 | 0.494/0.463 |
| | 336 | **0.515/0.467** | 0.622/0.515 | 0.519/0.474 | 0.544/0.492 |
| | 720 | 0.591/**0.509** | 0.644/0.50 | **0.582**/0.51 | 0.594/0.521 |
| | Avg | **0.501/0.458** | 0.575/0.481 | 0.506/0.466 | 0.515/0.475 |

Table 3: Ablation study on TEMPO.

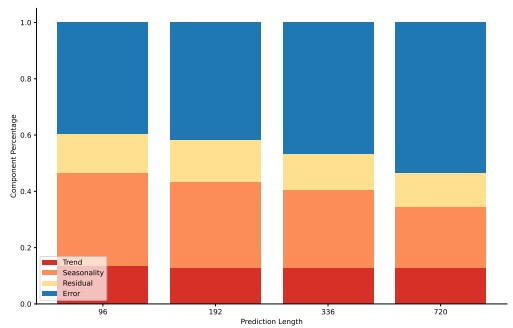

Figure 2: The SHAP values of decomposed components of TEMPO for ETTm1.

enhances the quality of individual decomposition components. This improvement in component quality is important for the model's forecasting accuracy. Note that employing the prompt design in isolation, without the support of decomposition, can detrimentally impact the backbone model's performance in most cases. This can be due to the difficulties in effectively prompting time series data from its raw form with limited semantic information. These findings underscore the essential nature of both prompt and decomposition elements in achieving robust forecasting capabilities under the zero-shot setting.

## 5.2 INTERPRETING MODEL PREDICTIONS

SHAP (SHapley Additive exPlanations) values serve as a comprehensive measure of feature importance, quantifying the average contribution of each feature to the prediction output across all possible feature combinations. As shown in Figure 2, when applied to our seasonal and trend decomposition, the SHAP values from the generalized additive model (GAM) suggest a dominant influence of the seasonal component on the model's predictions, implying a significant dependency of the model on the overall recurring patterns within the data. While the directional shifts of ETTm1 dataset's contribution is relatively stable. The escalating values in the 'Error' column, which denote the discrepancy between the model's predictions and the ground truth, indicate a potential decline in the model's accuracy as the prediction length increases which is indeed observed in most experiments run. In this context, the STL decomposition proves invaluable as it enables us to identify and quantify the individual contributions of each component to the overall predictions, as demonstrated by the SHAP values. This detailed understanding can yield critical insights in how the pre-trained transformer is interpreting and leveraging the decomposing pre-processing step, thereby providing a robust foundation for model optimization and enhancement. SHAP values for weather dataset can be found at Figure 14.

## 6 CONCLUSION

This paper proposes a soft prompt based generative transformer, TEMPO, which achieves state-of-the-art performance in zero-shot time series forecasting. We introduce the novel integration of prompts and seasonal trend decomposition together within a pre-trained Transformer-based backbone to allow the model to focus on appropriately utilizing knowledge from different temporal semantics components. Moreover, we demonstrate the effectiveness of TEMPO with multimodel input, effectively leveraging contextual information in time series forecasting. Lastly, with extensive experiments, we highlight the superiority of TEMPO in accuracy, and generalizability. One potential limitation worth further investigation is that superior LLMs with better numerical reasoning capabilities might yield better results. In addition, the encouraging results of TEMPO on the zero-shot experiments shed light into effective foundational models for time series.

## ACKNOWLEDGEMENT

This work is partially supported by the NSF Award #2125142 and NSF Award #2226087. The funding from these sources has been a cornerstone in enabling us to bring our project to fruition. We would like to extend our thanks to Yizhou Zhang, James Enouen, Qiang Huang, Chuizheng Meng, and Hao Niu for their invaluable discussions and insights in shaping the direction and execution of our work. We are also deeply grateful to the anonymous reviewers for their rigorous review process. Their detailed comments and constructive suggestions have significantly contributed to the improvement of this paper. The time and effort they invested in providing feedback have been invaluable and have greatly assisted us in refining our work.

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

# A  SHOWCASES

## A.1  COMPARE WITH GPT4TS

In Figure 3, 4, 5, 6, 7, we plot the comparison of the predicted value from our model and GPT4TS model given a look-back window. As shown in the datasets, we are able to predict close to the ground truth, which is also shown through our superior performance over other models in table 1. We select time series with different characteristics under different prediction lengths $O \in \{96, 192\}$: time series with high variability (Figure 5 a), periodic (Figure 3 a, Figure 3 b, 4 a, 4 b), non-periodic with a change in trend (Figure 6 a, Figure 6 b)

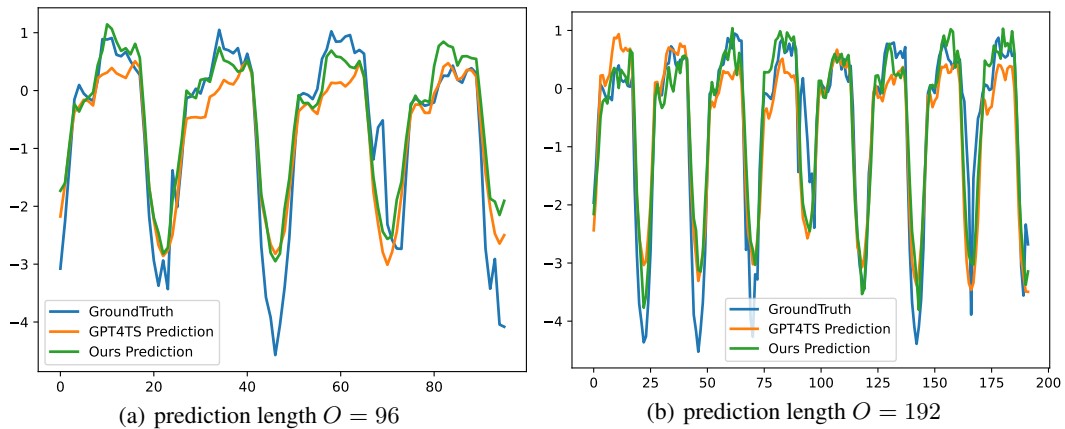

(a) prediction length $O = 96$    (b) prediction length $O = 192$

Figure 3: Visualization of long-term forecasting results. Compared between our model TEMPO and GPT4TS on ETTh1 dataset

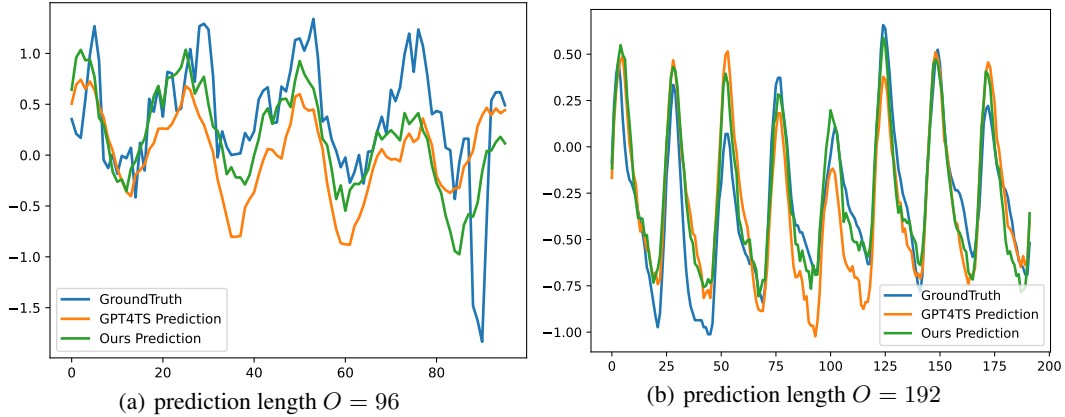

(a) prediction length $O = 96$    (b) prediction length $O = 192$

Figure 4: Visualization of long-term forecasting results. Compared between our model TEMPO and GPT4TS on ETTh2 dataset

## A.2  COMPARE WITH TIMEGPT

We also compare our results with TimeGPT (Garza & Mergenthaler-Canseco, 2023), which is capable of generating accurate predictions for a diverse range of datasets not seen during training, demonstrating superior performance in zero-shot inference compared to traditional statistical, machine learning, and deep learning methods. Access to TimeGPT-1 (Beta) is provided through a Python SDK and a REST API. This accessibility allows us to explore TimeGPT's forecasting capabilities on our datasets. As shown in Figure 8 and Figure 9, despite its design for various downstream tasks, it

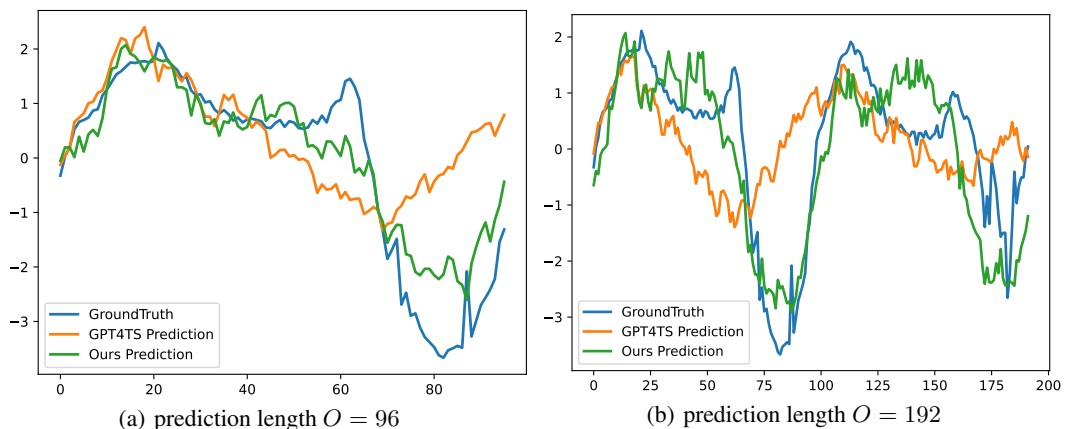

Figure 5: Visualization of long-term forecasting results. Compared between our model TEMPO and GPT4TS on ETTm1 dataset

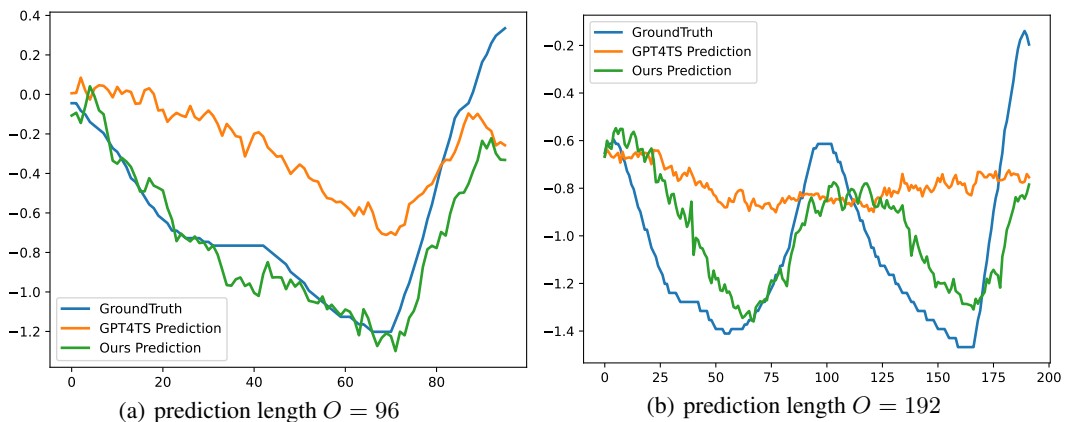

Figure 6: Visualization of long-term forecasting results. Compared between our model TEMPO and GPT4TS on ETTm2 dataset

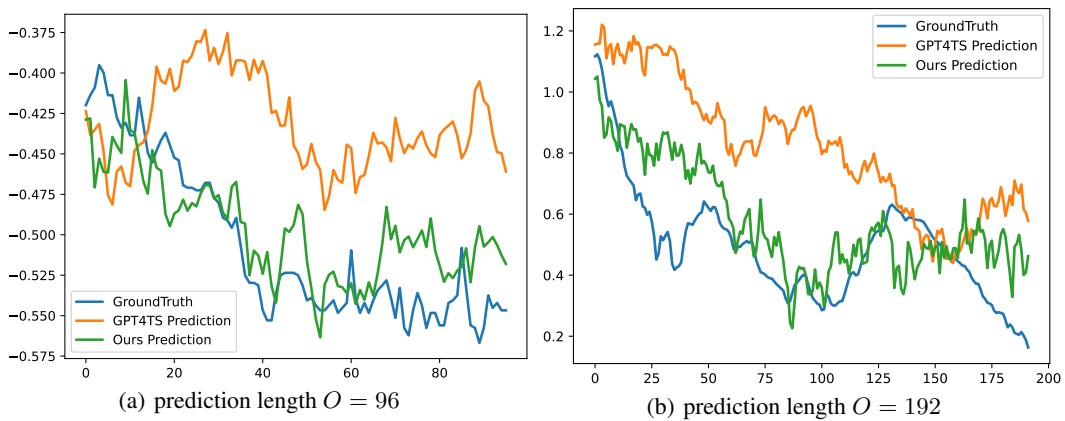

Figure 7: Visualization of long-term forecasting results on weather dataset. Compared between our model TEMPO and GPT4TS on weather dataset

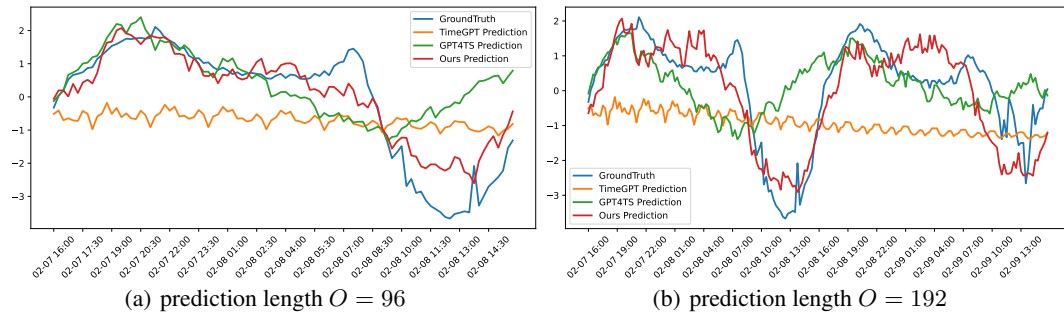

(a) prediction length $O = 96$      (b) prediction length $O = 192$

Figure 8: Visualization of long-term forecasting results on ETTm1 dataset. Compared between our model TEMPO and TimeGPT on weather dataset

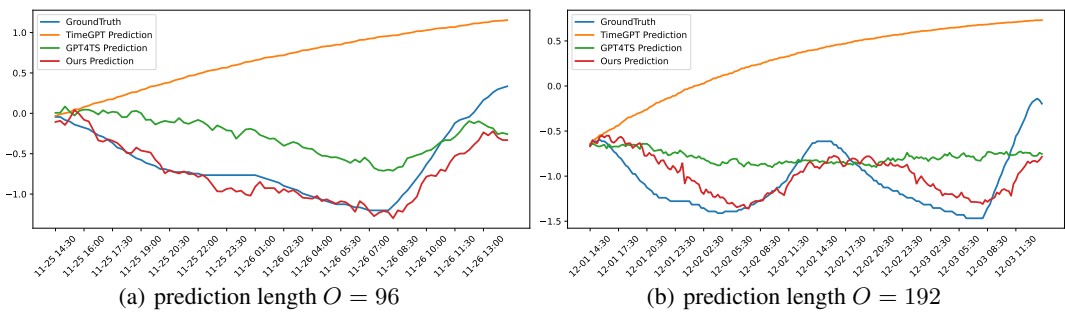

(a) prediction length $O = 96$      (b) prediction length $O = 192$

Figure 9: Visualization of long-term forecasting results on ETTm2 dataset. Compared between our model TEMPO and TimeGPT on weather dataset

is important to note that TimeGPT may not perform as well in long-term forecasting scenarios. In contrast, our proposed model excels in zero-shot settings, including long-term forecasting, illustrating the need for foundation models that can adapt to both the breadth of time series applications and the depth of forecasting horizons.

## B  EXPERIMENT SETTING

### B.1  TOWARDS FOUNDATION MODEL EXPERIMENTS DETAILS

Table 4: Dataset details of benchmark dataset.

| Dataset | Length | Covariates | Sampling Period |
|---|---|---|---|
| ETTh | 17420 | 7 | 1 hour |
| ETTm | 69680 | 7 | 15 min |
| Weather | 52696 | 22 | 10 min |
| Electricity | 26304 | 321 | 1 hour |
| Traffic | 17544 | 862 | 1 hour |

It has been well-established that channel-independence works well for time series datasets, so we treat each multivariate time series as multiple independent univariate time series. We use popular time series benchmark datasets (Zhou et al., 2021): ETTm1, ETTm2, ETTh1, ETTh2, Weather, Electricity, Traffic, ILI and exchnge. 1) ETTm1, ETTm2, ETTh1, ETTh2 contain electricity load from two electricity stations at 15 minutes level and hourly level. 2) Weather dataset contains 21 meteorological indicators of Germany within 1 year; 3) Electricity dataset contains electricity consumption; 4) Traffic dataset contains the occupation rate of the freeway system across the State of California. The lookback window $L$ is following (Zhou et al., 2023), and the prediction length $O$ is set to $\{96, 192, 336, 720\}$. In this experiment part, our experiments were conducted using single NVIDIA A100 GPU, with a batch size set to 256, and focused on long-term forecasting by employing a Mean Squared Error (MSE) loss function. To

ensure the reliability of our results, we performed three iterative loops and calculated the average of the outcomes. Our exploration covered [3, 6] GPT layers and tested various weights, [0.001, 0.01, and 1], for the MSE loss function applied to the reconstructed components of the time series. We have documented the optimal results obtained from this search. A comprehensive analysis of the impact that the number of GPT layers has on the performance will be addressed in future research.

**Towards Foundation Model's Zero Shot Setting** For each prediction length, we train a model on a mixture of training data from different domains and test the model on the target unseen domain's data. We construct the combined training dataset by pooling the training data and fullyshufflinge them. To prevent undue bias and ensure fair representation of data from each domain in the combined training data, we select an equal number of training examples from each domain's training data. We noted that the number of training samples that ETTh1 and ETTh2 has is on a much smaller magnitude compared to the other three training datasets (ETTm1, Weather, Electricity), so selecting the minimum number of training samples among all other training datasets would result in too much data loss from ETTm1, Weather, and Electricity, etc. Therefore, we included all training examples from ETTh1 and ETTh2 in the combined training dataset. Similar to traditional experimental settings, each time series (ETTh1, ETTh2, ETTm1, Weather, Electricity, ETTm2, Traffic) is split into three parts: training data, validation data, and test data following in 7:1:2 ratio in (Zhou et al., 2022), and we only merge the training and validation data. For ETTm1, ETTm2, Weather and Electricity data, the number of examples sampled to be pooled into the combined training dataset is chosen to be the minimum number of training examples among these training datasets.

## B.2 PROPOSED TETS DATASET SETTING

---

**Prompt**:
Suppose you are living in {Year: 2000}, can you help me summarize the news and reports in {Year: 2000}'s {quarter: second quarter} for {company name: Company A}, which is an {company sector: Technology} company. Please directly give me the answer limited to 2 sentences without apology.

**GPT Response:**
*Description:*
In the second quarter of 2000, Company A reported a net profit of $233 million, up from $123 million in the same quarter of the previous year, driven by strong sales of its X computers and Products Y. However, the company's stock price dropped after warning that its third-quarter profits would be below expectations due to slower sales.

---

Figure 10: Example for designing prompts using OPENAI ChatGPT-3.5 API.

**Data Collection** Our time series data for financial analysis and forecasting are derived primarily from the financial statements of companies including balance sheets, income statements, and cash flow statements. Specifically, we utilize data from the 500 largest U.S. companies across 11 sectors as listed in the Standard & Poor's 500 Index (S&P 500), which we divide into two parts: the first seven sectors for training and evaluation, and the remaining four for zero-shot forecasting tasks to test the model's ability to predict in unseen domains. While collecting corresponding contextual information from the abundance of digital news sources is challenging, OpenAI's ChatGPT API offers a solution to gather and condense relevant news efficiently. By inputting key details into the API and limiting the response to 110 tokens, as shown in Figure 10, we can swiftly extract pertinent contextual data to improve our analysis. Please refer to Section H for further details of creating TETS dataset.

**Prediction objective** The primary objective of our experiment is to forecast the Earnings Before Interest, Taxes, Depreciation and Amortization(EBITDA) for companies listed in S&P500, and our data range from 2000 to 2022. Following the multivariate time series framework presented in (Papadimitriou et al., 2020), we select foundational financial metrics from the income statements as input features: cost of goods sold (COGS), selling, general and administrative expenses (SG&A), RD expenses (RD_EXP), EBITDA, and Revenue. Comparing with other metrics, the selected metrics contain information more relevant to our prediction objective. For Large Language based models,

including our model TEMPO, GPT4TS, and T5, we apply channel-independence strategy to perform univariate time series forecasting tasks. All five features are used for training (predicting its future value based on its past value), while only EBITDA is accessible during the training stage. Other models follow the multivariate time series forecasting setting, treating the five features as multivariate input and predicting the target, EBITDA, both in the training and testing stages.

We predict quarterly EBITDA based on the past 20 quarters' data. This predicted value is then used to forecast the next quarter's EBITDA, iteratively four times, leading to a yearly prediction. In order to measure the accuracy of these predictions based on the cumulative yearly value (sum of 4 quarters), we employ the symmetric mean absolute percentage error (SMAPE) as the evaluation metric as well as the forecasting loss function in this experimental part.

**Data Split**  For companies under each sector, we employ the windowing method to generate cohesive training and testing instances. Under the channel-independence setting where we separate each feature to obtain univariate time series, we get 80,600 samples from the seven in-domain sectors, and 9,199 samples from the four zero-shot sectors(also known as cross-domain sectors), five as much as we get in the channel dependent setting. The sectors splitting is elaborated in H. In our experiments shown in table 2, We use 70% of in-domain data for training, 10% of in-domain data for evaluation, and all zero-shot data for unseen testing.

**Symmetric Mean Absolute Percentage Error**  In reality, the magnitude of financial metrics can vary significantly among different companies. So, we choose the symmetric mean absolute percentage error (SMAPE), a percentage-based accuracy measure, as our evaluation metric. For EBITDA, there are many negative results that may influence the final SMAPE. We use the form of SMAPE-Abs SMAPE:

$$\text{AbsSMAPE} = \frac{200\%}{n} \sum_{t=1}^{n} \frac{|F_t - A_t|}{|F_t| + |A_t|}, \tag{4}$$

Here, $F_t$ represents the true value, $A_t$ represents the predicted value in our system, and $n$ represents the total time steps we need to forecast.

SMAPE can be particularly sensitive to outliers. Specifically, when the true data and prediction have opposite signs, the resulting error may be up to 200%, seriously distorting the final results. Following the approach in (Papadimitriou et al., 2020), we filter out data points at the 80% and 90% thresholds and find most of the outliers are related to significant financial shifts due to mergers & acquisitions (M&A).

### B.3  GDELT DATASET SETTING

We utilized the GDELT dataset (Jia et al., 2024), which focuses on predicting the respective mentions and mentions in the news media. We utilized the data collected from the 55 regions under the US and the national data for the US and divided the 10 event root types in the dataset into unseen and seen sets, as demonstrated in Table 5. We focused on predicting the three key variables NumMentions, NumArticles, NumSources related to the particular event type within a given timeframe and geographical region. We apply channel-independence strategy to perform univariate time series forecasting tsks for all baseline models and our model. All three features are used for training and evaluation (predicting its future value based on its past value).

We predict the future 7 days based on the past 15 days' data directly. In other to measure the accuracy of the predictions, we use mean square error (MSE) and mean absolute error (MAE). For each region, we employ the windowing method to generate cohesive training and testing instances for each event root type. Under our channel-independence setting, we get 122,008 samples from the seven in-domain sectors (seen sectors) for training, and 76,048 samples for evaluating under the three zero-shot sectors (unseen sectors). In our experiments, we use 70% of in-domain data for training, 10% for evaluation and all zero-shot data for unseen testing.

| | Event Number | Event Type Name |
|---|---|---|
| | 01 | Make Public Statement |
| | 02 | Appeal |
| | 03 | Express Intent to Cooperate |
| Training Event | 04 | Consult |
| | 05 | Engage in Diplomatic Cooperation |
| | 07 | Provide Aid |
| | 08 | Yield |
| | 11 | Disapprove |
| Test Event | 17 | Coerce |
| | 19 | Fight |

Table 5: Event number to event type Name table

## C  FURTHER RESULTS

### C.1  SELF-SUPERVISED REPRESENTATION LEARNING

Our proposed model architecture can be designed to support self-supervised learning and thus further embrace foundation models for time series. Following  (Nie et al., 2023), we mask a random subset of patches by replacing them with zeros, where the patches are divided into non-overlapping patches for simplicity and to avoid masked patches influencing predictions. The prediction head is removed and replaced with a linear layer to reconstruct the masked patches. The model is trained to minimize the MSE between the predicted and true masked patches. To handle multivariate time series with varying numbers of features, we apply channel independence (Zeng et al., 2023) to model each time series independently.

With the strong performance TEMPO showed under the experiment 'many-to-one' zero-shot setting, from the perspective of a self-supervised cross-domain foundational model, we further investigate if using a TEMPO model trained on datasets across domains can still achieve comparable performance on unseen domains. Here, we still use the 'many-to-one' setting but the model is trained in a self-supervised manner. Specifically, we first use all other domain's data to train a representation model then only use 5% data of the training data to fine turn the total model with the prediction layer as a forecasting downstream task. Table 6 provides a comprehensive comparison of our model against other baseline models on three multivariate time series datasets that are unseen by the models during training, namely electricity and traffic and weather. All these selected 3 datasets are entirely dissimilar to any data the model has encountered before. TEMPO outperforms baseline models, achieving the lowest MSE and MAE in most cases. Note that TEMPO's average MSE and MAE is **7.3%** and **4.6%** less than the best-performing baseline model (GPT2) for the weather dataset, respectively. This finding shed light on the strong generalizability of TEMPO and indicated its potential of serving as a foundational time series forecasting model, maintaining robust performance for unseen domains.

### C.2  COMPARING WITH FULL-SHOT STATE-OF-THE-ARTS BASELINES

Towards foundation model training differs significantly from the one-to-one/many scenarios, where pre-training involves a homogenous dataset, often with consistent season patterns, sampling rates, and temporal scales. This homogeneity facilitates pattern learning transferable to fine-tuned datasets. In contrast, towards foundation model training involves pre-training on highly diverse datasets, such as merging traffic and weather data, which may hinder the model's ability to discern underlying patterns. In Table 7, we provide further results on ETTh1 and ETTh2 datasets, demonstrating that the performance of TEMPO (zero-shot setting) surpasses that of state-of-the-art models specifically designed for these target datasets with full-shot settings. The results in Table 7 are obtained from (Liu et al., 2023), including but not limited to iTransformer(Liu et al., 2023), Crossformer (Zhang & Yan, 2022), TiDE (Das et al., 2023a) and SCINet (Liu et al., 2022), which are also reported in our contemporaneous work, MOIRAI (Woo et al., 2024).

Table 6: Self-supervised representation learning results are fine-tuned on 5% in-domain datasets. We use prediction length $O \in \{96, 192, 336, 720\}$. A lower MSE indicates better performance, and the best results are in bold.

| | | TEMPO | GPT4TS | T54TS | Bert4TS | PatchTST |
|---|---|---|---|---|---|---|
| | | MSE/MAE | MSE/MAE | MSE/MAE | MSE/MAE | MSE/MAE |
| ECL | 96 | **0.19/0.29** | 0.202/0.301 | 0.199/0.293 | 0.202/0.298 | 0.21/0.308 |
| | 192 | **0.211/0.31** | 0.217/0.313 | 0.238/0.337 | 0.227/0.321 | 0.223/0.312 |
| | 336 | **0.229/0.323** | 0.258/0.353 | 0.273/0.364 | 0.256/0.345 | 0.282/0.357 |
| | 720 | **0.375/0.444** | 0.43/0.475 | 0.455/0.49 | 0.442/0.479 | 0.606/0.561 |
| | Avg | **0.251/0.342** | 0.277/0.361 | 0.291/0.371 | 0.282/0.361 | 0.33/0.385 |
| Traffic | 96 | 0.56/0.411 | 0.607/0.417 | **0.543**/0.408 | 0.591/0.423 | 0.577/**0.403** |
| | 192 | **0.575**/0.419 | 0.603/0.421 | 0.594/0.431 | 0.613/0.432 | 0.596/**0.411** |
| | 336 | **0.597/0.433** | 0.63/0.435 | 0.659/0.458 | 0.639/0.445 | 0.665/0.454 |
| | 720 | 0.65/0.452 | **0.643/0.439** | 0.69/0.49 | 0.744/0.496 | 0.802/0.501 |
| | Avg | **0.595**/0.429 | 0.621/**0.428** | 0.622/0.447 | 0.647/0.449 | 0.66/0.442 |
| Weather | 96 | **0.217/0.268** | 0.288/0.31 | 0.252/0.288 | 0.237/0.288 | 0.249/0.285 |
| | 192 | **0.265/0.302** | 0.305/0.331 | 0.322/0.336 | 0.291/0.323 | 0.277/0.314 |
| | 336 | 0.322/0.342 | 0.338/0.353 | 0.346/0.358 | 0.335/0.354 | **0.311/0.341** |
| | 720 | 0.41/0.397 | **0.381/0.377** | 0.444/0.42 | 0.466/0.436 | 0.385/0.386 |
| | Avg | **0.304/0.327** | 0.328/0.343 | 0.341/0.351 | 0.332/0.35 | 0.305/0.331 |

Table 7: Results of long term forecasting experiments on ETTh1 and ETTh2. The best results are marked in bold and the second optimal in underlined, respectively with MSE/MAE. Note that the TEMPO is under zero-shot setting and other models are under full-shot setting.

| | | TEMPO | iTransformer | TimesNet | PatchTST | Crossformer | TiDE | DLinear | SCINet | FEDformer |
|---|---|---|---|---|---|---|---|---|---|---|
| | | MSE/MAE | MSE/MAE | MSE/MAE | MSE/MAE | MSE/MAE | MSE/MAE | MSE/MAE | MSE/MAE | MSE/MAE |
| ETTh1 | 96 | 0.400/0.406 | 0.386/0.405 | 0.384/0.402 | 0.414/0.419 | 0.423/0.448 | 0.479/0.464 | 0.386/**0.400** | 0.654/0.599 | **0.376**/0.419 |
| | 192 | 0.426/**0.421** | 0.441/0.436 | 0.436/0.429 | 0.460/0.445 | 0.471/0.474 | 0.525/0.492 | 0.437/0.432 | 0.719/0.631 | **0.420**/0.448 |
| | 336 | **0.441/0.430** | 0.487/0.458 | 0.491/0.469 | 0.501/0.466 | 0.570/0.546 | 0.565/0.515 | 0.481/0.459 | 0.778/0.659 | 0.459/0.465 |
| | 720 | **0.443/0.451** | 0.503/0.491 | 0.521/0.500 | 0.500/0.488 | 0.653/0.621 | 0.594/0.558 | 0.519/0.516 | 0.836/0.699 | 0.506/0.507 |
| | Avg. | **0.428/0.427** | 0.454/0.447 | 0.458/0.450 | 0.469/0.454 | 0.529/0.522 | 0.541/0.507 | 0.456/0.452 | 0.747/0.647 | 0.440/0.460 |
| ETTh2 | 96 | 0.301/0.351 | **0.297**/0.349 | 0.340/0.374 | 0.302/**0.348** | 0.745/0.584 | 0.400/0.440 | 0.333/0.387 | 0.707/0.621 | 0.358/0.397 |
| | 192 | **0.355/0.389** | 0.380/0.400 | 0.402/0.414 | 0.388/0.400 | 0.877/0.656 | 0.528/0.509 | 0.477/0.476 | 0.860/0.689 | 0.429/0.439 |
| | 336 | **0.379/0.408** | 0.428/0.432 | 0.452/0.541 | 0.426/0.433 | 1.043/0.731 | 0.643/0.571 | 0.594/0.541 | 1.000/0.744 | 0.496/0.487 |
| | 720 | **0.409/0.440** | 0.427/0.445 | 0.462/0.657 | 0.431/0.446 | 1.104/0.763 | 0.874/0.679 | 0.831/0.657 | 1.249/0.838 | 0.463/0.474 |
| | Avg. | **0.361/0.398** | 0.383/0.407 | 0.414/0.427 | 0.387/0.407 | 0.942/0.684 | 0.611/0.550 | 0.559/0.515 | 0.954/0.723 | 0.437/0.449 |

Table 8: Compare the results with ARIMA.

| | ECL | | Traffic | | Weather | | Ettm2 | |
|---|---|---|---|---|---|---|---|---|
| | TEMPO | ARIMA | TEMPO | ARIMA | TEMPO | ARIMA | TEMPO | ARIMA |
| | MSE/MAE | MSE/MAE | MSE/MAE | MSE/MAE | MSE/MAE | MSE/MAE | MSE/MAE | MSE/MAE |
| 96 | **0.178/0.276** | 1.220/0.814 | **0.476/0.343** | 1.997/0.924 | **0.211/0.254** | 0.217/0.258 | **0.185/0.267** | 0.225/0.301 |
| 192 | **0.198/0.293** | 1.264/0.842 | **0.496/0.355** | 2.044/0.944 | **0.254/0.298** | 0.263/0.299 | **0.243/0.304** | 0.298/0.345 |
| 336 | **0.209/0.309** | 1.311/0.866 | **0.503/0.356** | 2.096/0.960 | **0.292/0.332** | 0.330/0.347 | **0.309/0.345** | 0.370/0.386 |
| 720 | **0.279/0.355** | 1.364/0.891 | **0.538/0.376** | 2.138/0.971 | **0.393/0.387** | 0.425/0.405 | **0.386/0.395** | 0.478/0.445 |
| Avg. | **0.216/0.308** | 1.290/0.853 | **0.503/0.357** | 2.069/0.950 | **0.287/0.318** | 0.309/0.327 | **0.280/0.328** | 0.343/0.369 |

### C.3 COMPARING WITH ARIMA

As a pioneering foundation model, TEMPO is engineered to forecast future values directly, eliminating the necessity for retraining with each new data instance. Its underlying framework captures intricate temporal patterns, granting it the versatility to generalize across various time series. In this study, we compare TEMPO's forecasting prowess with that of the ARIMA model (Hyndman & Khandakar, 2008), which is renowned for its capacity to make accurate predictions within a specific time series once the initial model parameters have been set. While ARIMA models excel in continuing predictions within the series they are configured for, they do not inherently possess the faculty to forecast across disparate time series without recalibration. We obtain the ARIMA's forecasting results from (Challu et al., 2023). As shown in Table 8, the results highlight the superior adaptability of our 'towards foundation model' – TEMPO – which retains its predictive accuracy even when applied to time series beyond its training scope, thereby illustrating the feasibility of more universal and resilient forecasting methodologies.

## D FURTHER ANALYSIS

### D.1 DESIGN OF PROMPT POOL

In this section, we propose another potential prompt design for addressing non-stationary nature of real-world time series data with distributional shifts (Huang et al., 2020). Specifically, we introduce a shared pool of prompts stored as distinct key-value pairs. Ideally, we want the model to leverage related past experiences, where similar input time series tend to retrieve the same group of prompts from the pool (Wang et al., 2022b). This would allow the model to selectively recall the most representative prompts at the level of individual time series instance input. In addition, this approach can enhance the modeling efficiency and predictive performance, as the model would be better equipped to recognize and apply learned patterns across diverse datasets via a shared representation pool. Prompts in the pool could encode temporal dependencies, trends, or seasonality effects relevant to different time periods. Specifically, the pool of prompt key-value pairs is defined as:

$$\mathbf{V_K} = \{(\boldsymbol{k}_1, V_1), (\boldsymbol{k}_2, V_2), \cdots, (\boldsymbol{k}_M, V_M)\}, \tag{5}$$

where $M$ is length of prompt pool, $V_m \in \mathbb{R}^{L_p \times L_E}$ is a single prompt with token length $L_p$ and the same embedding size $L_E$ as $\mathcal{P}_T^i$ and $\boldsymbol{k}_m \in \mathbf{K} = \{\boldsymbol{k}_m\}_{m=1}^M$ with the shape of $\mathbb{R}^{L_E}$. The score-matching process can be formulated with the score-matching function $\gamma\left(\mathcal{P}_T^i, \boldsymbol{k}_m\right) = \mathcal{P}_T^i \cdot \boldsymbol{k}_m / \|\mathcal{P}_T^i\| \|\boldsymbol{k}_m\|$, where $\gamma : \mathbb{R}^{L_E} \times \mathbb{R}^{L_E} \to \mathbb{R}$. The model is trained in an end-to-end way to optimize predictions with the prompts. The query $\mathcal{P}_T^i$ that is used to retrieve the top-$\mathcal{K}$ corresponding value comes from the patched time series input. Therefore, similar time series can be assigned to similar prompts. Denoting $\{s_j\}_{j=1}^{\mathcal{K}}$ as a subset of $\mathcal{K}$ indices for the selected top-$\mathcal{K}$ prompts, our input embedding of trend is as follows:

$$\boldsymbol{x}_T = [V_{s_1}; \cdots; V_{s_\mathcal{K}}; \mathcal{P}_T], \quad 1 \leq \mathcal{K} \leq M, \tag{6}$$

where we concatenate all the tokens along the temporal length dimension, so as $\boldsymbol{x}_S, \boldsymbol{x}_R$. Each instance can be assigned to multiple prompts, which can jointly encode knowledge pertinent to the forecasting task- such as periodic patterns exhibited by the time series, prevailing trends, or seasonality effects.

Table 9: Compare the different prompt designs on the ETTm2 dataset.

|  | Semi-soft mse/mae | Soft mse/mae | Hard mse/mae | Pool mse/mae | Pool mask all mse/mae |
|---|---|---|---|---|---|
| 96 | **0.182/0.263** | 0.189/0.271 | 0.182/0.267 | 0.185/0.267 | 0.1952/0.274 |
| 192 | 0.243/**0.304** | 0.252/0.307 | 0.243/0.302 | **0.242/0.304** | 0.2739/0.324 |
| 336 | 0.309/0.344 | 0.306/0.348 | 0.299/0.340 | **0.289/0.336** | 0.3131/0.354 |
| 720 | 0.384/0.392 | 0.386/0.394 | 0.380/0.392 | **0.373/0.386** | 0.3794/0.390 |
| Avg. | 0.280/0.326 | 0.283/0.330 | 0.276/0.325 | **0.273/0.323** | 0.290/0.335 |

## D.2 RESULTS ON DIFFERENT PROMPT DESIGN

In this section, we examine the impact of various prompt designs on model performance. We utilize the 'semi-soft' prompt as outlined in Section 3.3, where the prompt vectors are initialized semi-softly; the soft prompt, which entails the random initialization of vectors of identical dimensions to the 'semi-soft' prompt; and the hard prompt, which is semantically meaningful and remains fixed post-tokenization. Additionally, we explore the prompt pool, as described in Section D.1, and employ a similar leave-one-out approach to mask all prompts within the pool to investigate its effectiveness.

The findings, presented in Table 9, reveal that, in the ETTm2 dataset, the prompt pool outperforms the 'semi-soft' prompt in three out of four scenarios, underscoring the potential of prompts to enhance model capacity and adaptability to shifts in data distribution. Furthermore, we observe that prompts with explicit semantic content (Semi-soft and Hard) surpass the performance of simple soft prompts. This suggests that incorporating semantic information as discrete indicators within a pre-trained model can more effectively orchestrate domain knowledge. This understanding informs the design of prompts for efficient interaction with language models, especially in applications where precision and relevance of the output are crucial.

## D.3 ANALYSIS ON PROMPT POOL

Here is a summary of how the prompts are initialized and trained in our work:

- Initialization: The prompt embeddings in the pool are randomly initialized from a normal distribution, as is standard practice for trainable parameters in neural networks.
- Training: The prompts' value and all other model parameters are trained in an end-to-end manner to optimize the forecasting objective. This allows the prompts to be continuously updated to encode relevant temporal knowledge.

The number of prompts and embedding dimensions are treated as hyperparameters and tuned for good performance. Different pool settings, including pool size, top k number, and prompt length, will lead to different results. To explore this, we conduct a total of 27 experiments, setting 3 distinct values for each of the 3 settings: (1) pool size of 10, 20, and 30. (2) top $k$ numbers of 1, 2, and 3. (3) prompt lengths of 1, 2, and 3. We choose the combination with the best results for TEMPO settings. For the long-term and short-term forecasting experiments, we choose a pool size with $M = 30$ and $\mathcal{K}=3$ and prompt length is 3. Detailed design analysis provides insights into prompt similarity and selection. Note that, the prompt pool's key in (Wang et al., 2022b) is trainable which allows us to maintain consistent and distinct characteristics of time series data for analysis. Our work offers an initial exploration into prompt-based tuning for time series forecasting, but substantial room remains for advancing prompt pool design.

### D.3.1 PROMPT SELECTION DISTRIBUTION

To elucidate the mechanics behind prompt selection, we have visualized the distribution histograms for chosen prompts corresponding to the trend, seasonal, and residual elements of the ETTm2 dataset in Figure 11. In our experimental framework, each data point is permitted to select multiple prompts—with three prompts being chosen per component. Consequently, the frequency is determined by the number of times a particular prompt is selected across the dataset. The histograms reveal pronounced discrepancies in prompt preferences between periodic and seasonal components.

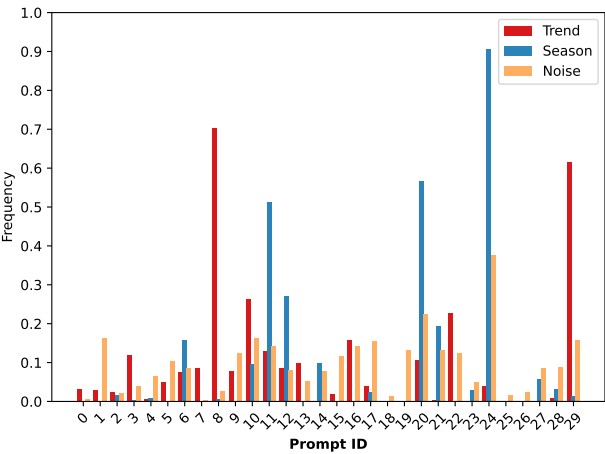

Figure 11: Prompt Distribution for prompt pool selection.

For instance, within the ETTm2 dataset, prompts 11, 20, and 24 are predominantly selected for capturing trends, whereas prompts 8, 10, and 29 are primarily chosen for seasonal fluctuations. This observation substantiates the premise that the prompt pool is adept at furnishing specialized prompts tailored to discrete characteristics of time series data.

## D.4 ANALYSIS ON DESIGNS OF INJECTING CONTEXTUAL INFORMATION

Table 10: SMAPE results of GEBDIT dataset with different textual information injection.

| | Event | Sum + TP | SumP + TP | Sum ⊕ TP | SumP ⊕ TP | Sum + Semi | SumP + Semi | Sum ⊕ Semi | SumP ⊕ Semi |
|---|---|---|---|---|---|---|---|---|---|
| SMAPE | 11 | 38.77 | 38.77 | **38.75** | 38.90 | 38.91 | 38.82 | 39.04 | 38.79 |
| | 17 | 41.02 | 41.03 | **40.95** | 41.05 | 41.24 | 41.08 | 41.38 | 41.08 |
| | 19 | 44.03 | 44.02 | **44.06** | 44.10 | 44.41 | 44.19 | 44.73 | 44.24 |

In this section, we investigate the influence of various configurations of textual injection and original prompt design from multi-modality perspective. As depicted in Table 10, eight distinct prompt designs were formulated. 'Sum' denotes the utilization of a direct summary of textual data as a prompt, while 'SumP' signifies the use of textual information as a query within the prompt pool. The symbols '+' and '⊕' represent summation and concatenation operations, respectively. 'TP' stands for 'time series prompt pool,' and 'Semi' indicates a 'semi-soft prompt' where we manually design the prompt, with trainable parameters, referred to as "Predict the future time step given the {*time series data type*}" for 3 different time series (Trend, Season, Residual) after decomposition. Each design choice exerts a distinct impact on the performance metrics. The direct incorporation of textual information along with the prompt pool yields the most optimal and near-optimal outcomes. In future work, we aim to delve deeper into the analysis of multimodal solution design strategies for time series forecasting.

## D.5 HIDDEN REPRESENTATION

Figure 12 demonstrates the difference between the representation of the output hidden space from the pre-trained langauge model. While the representation of time series learned from GPT4TS is centered as a whole, the representation of the decomposed component from TEMPO implies a certain soft boundary between the three components. This is a demonstration of how TEMPO is able to learn the representation of trend, seasonality, residual parts respectively, which contributes to the superior performance of our model TEMPO.

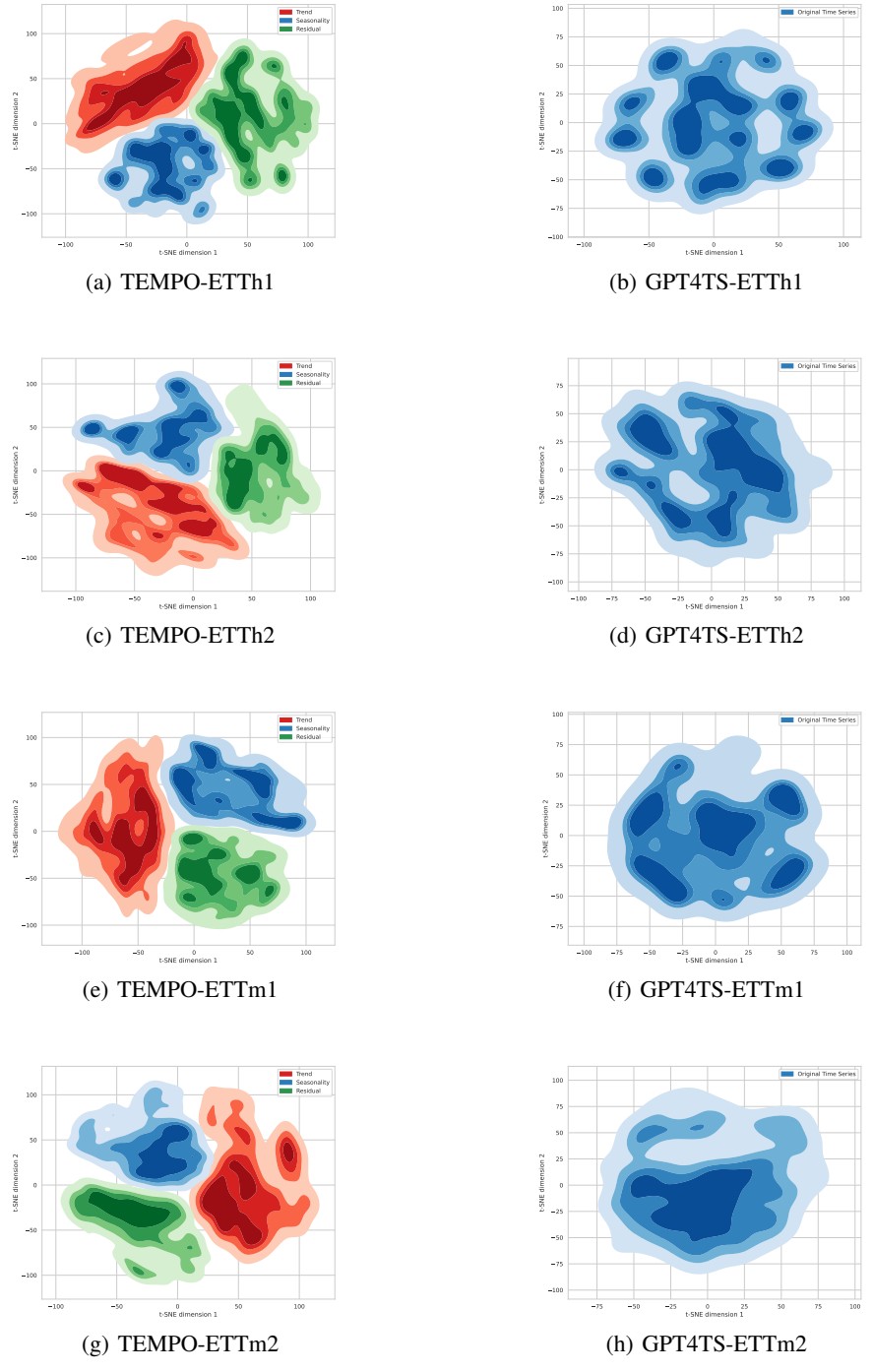

Figure 12: Comparison of GPT4TS representation with TEMPO representation for prediction length $O = 96$ using TSNE. Trend in red, seasonality in blue, residual in green.

### D.6 Model Training Time Comparison

Figure 13 illustrates the training time of other baseline models in comparison to our model TEMPO. To ensure fairness, we calculated the percentage of runtime for models operating on identical machines and utilizing equivalent computational resources. Each model's training time is presented as a ratio relative to TEMPO's training time. A value less than 1 indicates that the model trains faster than TEMPO, while a value greater than 1 suggests the opposite. We use horizontal bars to visually represent each model's relative training time, with the bars extending to the left or right of the central vertical line based on whether they are faster or slower than our model TEMPO, respectively.

Figure 13: Visual Comparison on relative training time of other models and our proposed model TEMPO under channel independent setting.

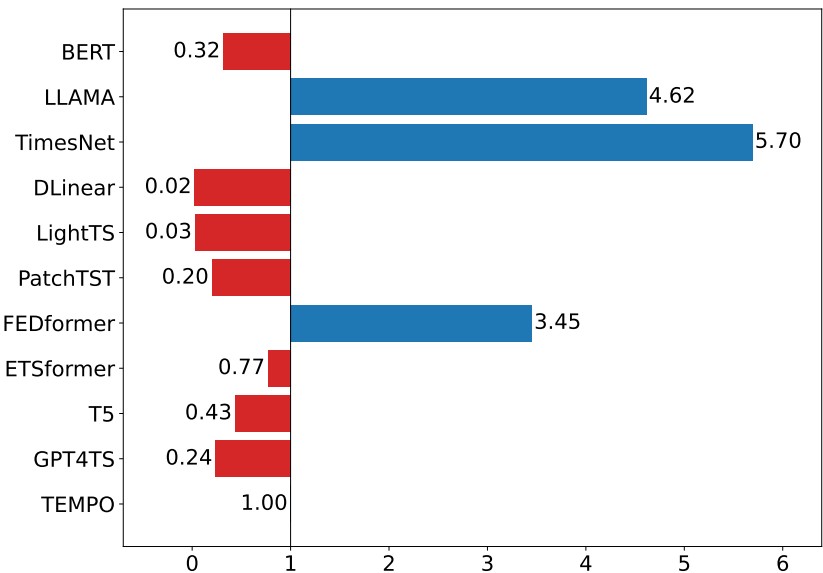

## E The roles of Generalized Additive Models (GAM) and SHapley Additive exPlanations (SHAP)

In our paper, GAM and SHAP serve as instrumental tools, not only for affirming anticipated findings but also for yielding deeper insights and explanations into the inner workings of intricate models.

- Role of GAM: GAM inherently models the effects of different features as additive components. This characteristic of GAM provides intrinsic interpretability to TEMPO. It's not merely a tool for confirming the absence of patterns in residuals; it also helps us understand how each feature contributes to the final prediction.
- Role of SHAP: SHAP helps in attributing feature effects post-hoc to explain the predictions made by complex models, which may otherwise be opaque.

The utility of GAM and SHAP in our analysis can be detailed as follows:

- Confirmation of Assumptions: the analyses quantitatively confirm assumptions about model behavior with data-driven evidence, rather than just intuition. This substantiation increases the trust and transparency in the model's predictions;
- Detecting Unexpected Behaviors: the component attribution could reveal unexpected behaviors if present. For example, residual impact being higher than expected could indicate overfitting noise.
- Providing Nuanced Insights: SHAP provides nuance beyond high-level expectations, like showing the increasing error of seasonal components in longer forecasts.

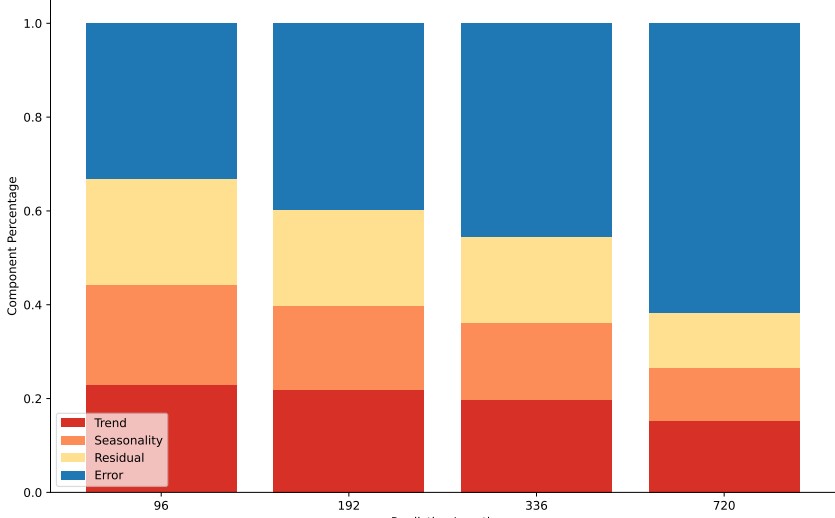

Figure 14: The SHAP (SHapley Additive exPlanations) values of decomposed components of TEMPO for weather dataset.

In our paper, we use the ETTm1 and weather datasets as detailed examples. The full results used to calculate the SHAP value can be found at Table 11. In datasets exhibiting strong seasonality, the seasonal component may display much larger variations than the residual component. Conversely, in datasets with minimal seasonality, the variations between these two components should be more comparable. We can calculate the strength of seasonality via:

$$S = \max\left(0, 1 - \frac{\mathrm{Var}\left(R_t\right)}{\mathrm{Var}\left(S_t\right) + \mathrm{Var}\left(R_t\right)}\right) \qquad (7)$$

When we compare the seasonality strengths of different datasets, we find that ETTm1 (as shown in Figure 2, with a seasonality strength of 0.99) constitutes strongly seasonal data, whereas the weather dataset (depicted in Figure 14 with a seasonality strength of 0.476) exhibits less seasonality and a more pronounced trend. These findings align with the conclusions drawn from the SHAP values. The performance degradation of ETTm1, when the prediction length is increased, can be primarily attributed to inaccuracies in the prediction of seasonal terms. In summary, SHAP provides pivotal descriptive power for model transparency, moving beyond intuition. The ability to discern how much and where components contribute enables targeted improvements. These insights can guide us in better leveraging inductive bias to enhance both efficiency and effectiveness in the era of pre-training models. One of the interesting future works is that we can adaptively and selectively optimize specific components based on the GAM structure and SHAP scores during the training process. This approach would allow us to focus our computational resources and efforts on the most influential components, thereby improving the overall effectiveness of the model.

## F    BASELINE MODEL EXPLANATIONS

We demonstrate the baseline models we compared with in our experiments in the following:

- DLinear (Zeng et al., 2023): DLinear combines a decomposition scheme from Autoformer and FEDformer with linear layers to predict time series data by modeling trend and seasonal components separately and summing their features for enhanced performance in trend-rich datasets.
- PatchTST (Nie et al., 2023): PatchTST is a Transformer-based model for multivariate time series forecasting that segments data into subseries patches and uses a channel-independent design to efficiently reduce computational costs while enhancing long-term prediction accuracy.

Table 11: SHAP original values for each component

| | | w/o trend | w/o season | w/o residual | trend | season | residual | empty set | complete set |
|---|---|---|---|---|---|---|---|---|---|
| | | MSE/MAE | MSE/MAE | MSE/MAE | MSE/MAE | MSE/MAE | MSE/MAE | MSE/MAE | MSE/MAE |
| ETTm1 | 96 | **0.437**/0.432 | 0.670/0.526 | 0.457/0.441 | 0.663/0.541 | 0.472/0.455 | 0.680/0.535 | 1.104/0.790 | 0.438/**0.424** |
| | 192 | 0.466/0.447 | 0.646/0.518 | 0.488/0.455 | 0.682/0.529 | 0.483/0.455 | 0.666/0.526 | 1.101/0.789 | **0.461/0.432** |
| | 336 | **0.505/0.466** | 0.672/0.530 | 0.526/0.476 | 0.680/0.531 | 0.524/0.475 | 0.707/0.543 | 1.102/0.790 | 0.515/0.467 |
| | 720 | **0.579/0.507** | 0.678/0.549 | 0.586/0.508 | 0.684/0.548 | 0.592/0.509 | 0.709/0.558 | 1.105/0.794 | 0.591/0.509 |
| | Avg | **0.497**/0.463 | 0.666/0.531 | 0.514/0.470 | 0.677/0.537 | 0.518/0.474 | 0.691/0.540 | 1.103/0.791 | 0.501/**0.458** |
| Weather | 96 | 0.213/0.267 | **0.202**/0.261 | 0.205/0.264 | 0.223/0.289 | 0.234/0.293 | 0.220/0.284 | 0.637/0.608 | 0.211/**0.254** |
| | 192 | 0.266/0.317 | **0.251/0.297** | 0.256/0.306 | 0.254/0.304 | 0.290/0.335 | 0.262/0.316 | 0.638/0.608 | 0.254/0.298 |
| | 336 | 0.317/0.356 | **0.290**/0.333 | 0.295/0.331 | 0.293/**0.331** | 0.328/0.357 | 0.313/0.356 | 0.640/0.609 | 0.292/0.332 |
| | 720 | 0.402/0.401 | 0.371/0.383 | 0.377/0.380 | 0.364/**0.378** | 0.389/0.393 | 0.385/0.398 | 0.638/0.610 | **0.370**/0.379 |
| | Avg | 0.300/0.335 | **0.279**/0.318 | 0.283/0.320 | 0.283/0.325 | 0.310/0.345 | 0.295/0.339 | 0.638/0.609 | 0.282/**0.316** |

- **FEDformer** (Zhou et al., 2022): FEDformer combines seasonal-trend decomposition with Transformers for time series forecasting, leveraging frequency insights for efficiency and accuracy, outperforming state-of-the-art methods.

- **Informer** (Zhou et al., 2021): Informer is a transformer-based model optimized for long sequence time-series forecasting, leveraging ProbSparse self-attention for efficiency, self-attention distilling for handling long inputs, and a generative decoder for rapid predictions.

- **ETSformer** (Woo et al., 2022): ETSformer is a novel Transformer architecture for time-series forecasting that integrates exponential smoothing principles, replacing traditional self-attention with exponential smoothing attention and frequency attention, to enhance accuracy, efficiency, and interpretability.

- **TimesNet** (Wu et al., 2023): TimesNet transforms 1D time series into 2D tensors capturing intra- and inter-period variations and uses TimesBlock with an inception block to extract complex temporal patterns, excelling in multiple time series tasks.

- **GPT-2** (Radford et al., 2019): GPT-2 is a decoder-based language model developed by OpenAI, designed to generate coherent and diverse textual content from a given prompt. In our work, we use the GPT-2 with 6 layers as the backbone, which is adapted from GPT4TS (Zhou et al., 2023).

- **BERT** (Devlin et al., 2019): BERT (Bidirectional Encoder Representations from Transformers) is an encoder-based deep learning model utilizing the Transformer architecture designed by Google to understand the context of words in a sentence by analyzing text bi-directionally.

- **T5** (Raffel et al., 2020): T5 (Text-to-Text Transfer Transformer) is a state-of-the-art neural network model with encoder-decoder based architecture designed by Google that converts every language problem into a text-to-text format.

- **LLaMA** (Touvron et al., 2023): LLaMA (Large Langauge Model Meta AI) is a collection of state-of-the-art foundation language models ranging from 7B to 65B parameters delivering exceptional performance, while significantly reducing the needed computational power and resources. In our work, we use the first 6 layers of 7B LLaMA.

# G  THEORICAL ANALYSIS

## G.1  PROOF OF THEOREM 3.1

**Theorem G.1** *Suppose that we have time series signal $Y(t) = S(t) + T(t) + R(t), t \in [t_1, t_n]$, where $S(t)$ is the seasonal signal (periodical), $T(t)$ is the trend signal (non-periodical) and $R(t)$ is the residual signal. Let $E = \{e_1, e_2, ..., e_n\}$ denote a set of orthogonal bases. Let $E_S \subseteq E$ denote the subset of $E$ on which $S(t)$ has non-zero eigenvalues and $E_T \subseteq E$ denote the subset of $E$ on which $T(t)$ has non-zero eigenvalues. If $S(t)$ and $T(t)$ are not orthogonal, i.e. $\sum_{i=1}^{n} S(t_i)T(t_i) \neq 0$, then $E_T \cap E_S \neq \emptyset$, i.e. $E$ can not disentangle the two signals onto two disjoint sets of bases.*

**Proof 1** *We decompose $S(t)$ and $T(t)$ onto $E$ and acquire that $S(t) = \sum a_i e_i$ and $T(t) = \sum b_i e_i$. Then it is obvious that $e_i \in E_S \iff a_i \neq 0$ and $e_i \in E_T \iff b_i \neq 0$. Now, let us consider the inner product of $S(t)$ and $T(t)$:*

$$\sum_{i=1}^{n} S(t_i)T(t_i) = S(t) \cdot T(t) = (\sum a_i e_i) \cdot (\sum b_i e_i) = \sum_{i,j} a_i b_j e_i e_j \tag{8}$$

*Note that the components found by PCA is a set of orthogonal basis. Thus, for any $i \neq j$, we have $e_i e_j = 0$. Thus, we have:*

$$\sum_{i=1}^{n} S(t_i)T(t_i) = S(t) \cdot T(t) = (\sum a_i e_i) \cdot (\sum b_i e_i) = \sum_i a_i b_i ||e_i||_2^2 \tag{9}$$

*Note that $\sum_{i=1}^{n} S(t_i)T(t_i) = 0$. Thus, there must be at least one $i$ such that $a_i \neq 0$ and $b_i \neq 0$. Thus, $e_i \in E_S$ and $e_i \in E_T$, in other words, $E_T \cap E_S \neq \emptyset$.*

The above theorem proves that if $T(t)$ and $S(t)$ are not orthogonal, then there does not exist a set of orthogonal bases that disentangle $S(t)$ and $T(t)$ onto two disjoint sets of bases. Note that it is common that a periodical signal is not orthogonal with a non-periodical signal. This is because the spectrum of a periodical signal is discrete and the spectrum of a periodical signal is continuous. Thus, it is very likely that there exist overlaps on those non-zero frequencies of the periodical signal. Note that PCA also aims at learning a set of orthogonal bases on the data. We can quickly acquire a corollary that PCA can not disentangle the two signals into two disjoint sets of bases. Based on (Zhou et al., 2023)'s Theorem 1, we can reveal that self-attention in pre-trained large models learns to perform a function closely related to PCA. Therefore, the self-attention mechanism cannot automatically decompose the time series into its trend and seasonal components unless we manually perform this operation.

## G.2 INTERPRETING MODEL PREDICTIONS FROM FREQUENCY DOMAIN

In addition to Section 5.2, which gives an experimental perspective on why decomposition can aid forecasting results, we provide a theoretical analysis from the spectral domain. Specifically, time series signals can be represented as a combination of different frequencies in the spectral domain. Forecasting is challenging because real-world series comprises convoluted mixtures of variations with overlapping periodicities. However, by shifting our view to the frequency domain, we can identify distinct components via STL decomposition containing isolated frequencies that stand out clearly from the rest of the spectrum. This separation of dominant periodic patterns is crucial because forecasting future values equates to predicting how these underlying frequencies evolve over time:

**Proposition G.2 (Equivalence of time domain forecasting and frequency domain forecasting )** *Assume $x_0, x_1, ..., x_{N-1}$ and $\hat{x}_0, \hat{x}_1 ..., \hat{x}_{N-1}, \hat{x}_N$ are the input and output sequences of the frequency model. Then, $\hat{x}_N$ transferred from the frequency domain is the predicted value at timestamp $N$.*

Given input sequence $\{x_t | t = 0, 1, ..., N - 1\}$, where $N$ is the number of discrete timestamps, in the time domain, the Discrete Fourier Transform (DFT, $F$) and inverse Discrete Fourier Transform (iDFT, $f$) operation to obtain the frequency domain can be defined as:

$$F(u) = \frac{1}{N} \sum_{x=0}^{N-1} f(x) e^{\frac{-i2\pi ux}{N}}, u = 0, 1, \ldots, N - 1, \tag{10}$$

$$f(x) = \sum_{u=0}^{N-1} F(u) e^{\frac{i2\pi ux}{N}}, x = 0, 1, \ldots, N - 1. \tag{11}$$

According to Proposition G.2, assuming that the next value of $F(u)$, can be predicted as $F'(N)$, other unknown variables in the time and frequency domains, including the $(N + 1)$th discrete sample $f(N)$ and the new DFT's result $F'(u), u = 0, 1, 2, \ldots, N - 1$ are determined by the given $F'(N)$.

**Proof 2** *Let*

$$A = \sum_{x=0}^{N-1} f(x)\left(\frac{e^{-\frac{i2\pi ux}{N}}}{N} - \frac{e^{-\frac{i2\pi ux}{N+1}}}{N+1}\right), \tag{12}$$

$$B = \frac{1}{N+1} \sum_{x=0}^{N-1} f(x)e^{-\frac{i2\pi Nx}{N+1}}, \tag{13}$$

*then we have:*

$$f(N) = (N+1)(F'(N) - B)e^{-\frac{i2\pi N^2}{N+1}}, \tag{14}$$

$$F'(u) = A + (F'(N) - B)e^{\frac{i2\pi(N-u)N}{N+1}}. \tag{15}$$

*For $u = 0, 1, 2, ..., N-1$, the value of $F'(u) - F(u)$ can be represented as:*

$$F'(u) - F(u) = A + \frac{1}{N+1}f(N)e^{-\frac{i2\pi uN}{N+1}}. \tag{16}$$

*For $u = N$, the value of $F'(N)$ can be represented as*

$$F'(N) = B + \frac{1}{N+1}f(N)e^{-\frac{i2\pi N^2}{N+1}} \tag{17}$$

.

*Given $F'(N)$, we can inference $F'(u)$ by:*

$$F'(u) = A + (F'(N) - B)e^{\frac{i2\pi(N-u)N}{N+1}}, u = 0, 1, 2, ..., N-1. \tag{18}$$

*and $f(N)$ by:*

$$f(N) = (N+1)(F'(N) - B)e^{-\frac{i2\pi N^2}{N+1}}, \tag{19}$$

*Thus, the only variable that needs to be predicted is $F'(N)$.*

This proposition reveals that if it is easy to predict patterns in the frequency domain, we can more easily predict the time series' future values. Forecasting equates to predicting the evolution of the underlying frequencies that make up the time series signal. STL decomposition significantly aids this task by separating components with distinct dominant periodic patterns. With STL, each component presents far fewer intertwining periodic influences to disentangle, which notably simplifies the prediction problem. For instance, the trend component may exhibit a lone annual cycle that clearly dominates its spectrum. A targeted predictive model focusing solely on accurately estimating the progression of this isolated frequency can generate accurate forecasts. Likewise, the seasonal element neatly isolates recurring daily or weekly frequencies. Models tailored specifically for these known periodicities allow for highly predictable extrapolations. In contrast, directly modeling the raw data's condensed spectrum with numerous blended periodic components yields unsatisfactory approximations. The overlapping frequencies are difficult to distinguish and predict independently.

Conceptualizing forecasting through a frequency domain lens reveals how STL decomposes complex spectral mixtures into distinguishable frequency-based sub-problems. This allows implementation optimized predictive strategies to uncover patterns in each component for markedly improved time series predictions. In essence, STL facilitates accurate future predictions by disentangling the spectral content into simpler predictable forms.

## H    DETAIL OF THE TETS DATASET

**Time series data**    Analyzing and forecasting a company's future profitability and viability are essential for its development and investment strategies. Financial assessment and prediction are data-driven, mostly relying on the combination of diverse data types including company reports, etc. In this project, our primary sources are the company's financial statements: balanced sheet, income statements, and cash flow statements.

The Standard & Poor's 500 Index (S&P 500) represents a stock market index that measures the stock performance of the 500 largest companies in the U.S.11 sectors in the S&P500 are included

in our dataset: Basic Materials (21 companies), Communication Services (26 companies), Energy (22 companies), Financial Services (69 companies), Healthcare (65 companies), Technology (71 companies), Utilities (30 companies), Consumer Cyclical (58 companies), Consumer Defensive (36 companies), Industrials (73 companies), Real Estate (32 companies). In terms of dataset division, we separate the sectors in our dataset to achieve both in-domain task setting and zero-shot task setting. The first seven sectors are treated as training and evaluation sectors, while the last four sectors are reserved as unseen sectors for zero-shot forecasting task.

To address missing numerical information for companies in the S&P 500 that lack data prior to 2010, we apply linear interpolation after experimenting with various methods. Linear interpolation is a technique that estimates a value within a range using two known end-point values. For missing values in research and development expenses, we adopted a zero-filling strategy. This is because null entries in these statements typically indicate that the company did not make any investment in that area.

**Contextual data collection**  This rise of Large-scale pre-trained models (LLMs) in the field of Natural Langauge Processing has provided new possibilities for their application in time seris analysis. LLMs have proven useful for analyzing and learning complicated relationships and making inferences across different time series sequences. However, most existing approaches primarily convert time series data to direct input into LLMs, overlooking the fact that the LLMs are pre-trained specifically for natural language and thus neglecting the incorporation of contextual data.

Further, the information contained in time series data is limited, especially in the financial field. Time series data in the financial field, such as company statements, primarily reflect the financial numeric changes based on the company's historical strategy and broader macroeconomic shifts. These data contain the company's internal historical information. However, the broader market environment, referred to as external information, also plays an important role in the company's future development. For example, medicine and healthcare companies experienced steady growth before the outbreak of COVID-19. But between 2019 and 2020, after the outbreak of the pandemic, the financial statements of such companies were impacted significantly. As a result, we recognize the value of integrating news and reports as external data sources to complement internal information contained in time series data. The information contained in the external data mainly includes 3 parts: (i). Policy shifts across regions (ii). Significant events occurring globally (iii). Public reaction to companies' products. Together, these elements provide supplementary information missing in time series data (internal data), therefore enhancing our forecasting capabilities.

Extracting contextual data, such as news and reports, from varied sources presents a significant challenge. In today's digital age, numerous news websites and apps deliver a wide range of world news, spanning from influential news affecting entire industries to trivial, minor reports. Thus, it is crucial to filter and summarize the information, distinguishing between pivotal and less significant news. Fortunately, the recently released ChatGPT API[2] by Open AI offers the capability of collecting and summarizing news and reports for a specified duration.

Through consolidating all relevant details – query, quarter, yearly context, company information, and specific requirements – into user message and setting a cap at 110 tokens for response, we can efficiently obtain the desired contextual information from ChatGPT API. For illustration, Figure 10 displays an example from company A, showcasing designed prompts and corresponding responses from ChatGPT 3.5. If the contextual information can not be generated, the API often returns messages with keywords such as 'unfortunately' and 'sorry'. We detect and replace them with the term 'None', representing neutral contextual information. Additionally, Figure 15 and 17 provide a illustration of our dataset, encompassing both time series data and the corresponding contextual texts. A detailed view of the contextual texts can be seen in Figure 16 and 18.

---

[2]https://platform.openai.com/docs/guides/gpt

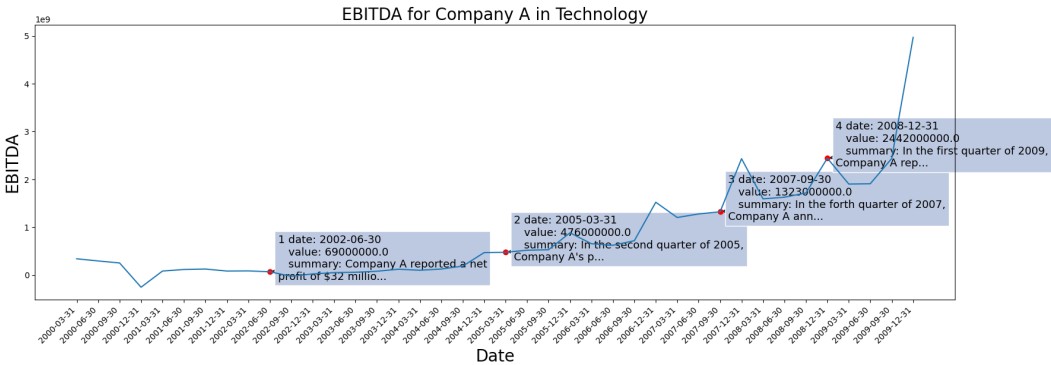

Figure 15: EBITDA for Company A with contextual information

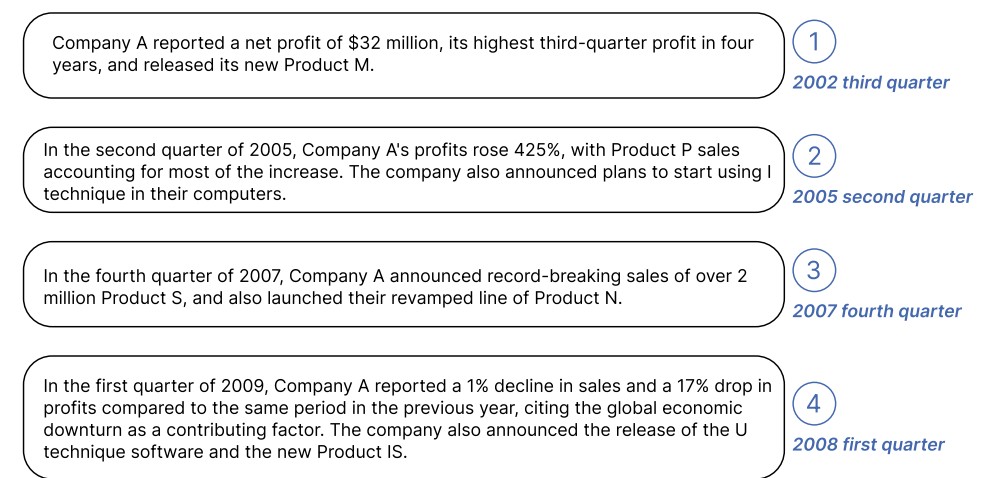

Figure 16: Example of generated contextual information for Company A marked in Figure 15

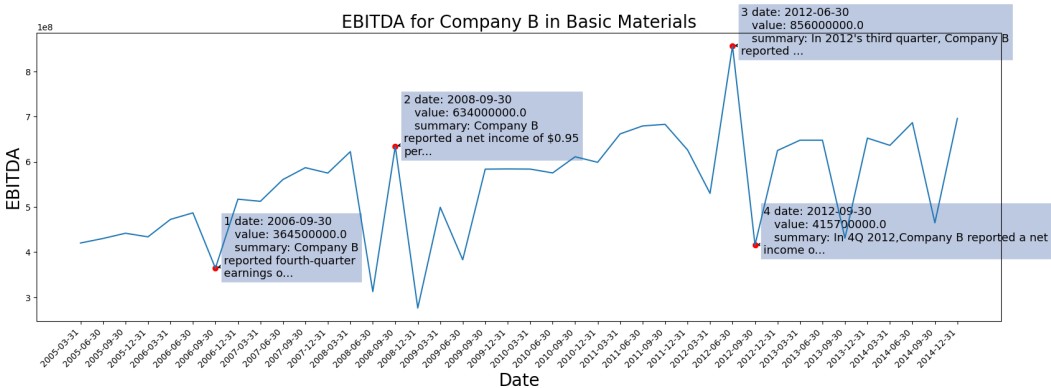

Figure 17: EBITDA for Company B with contextual informatino

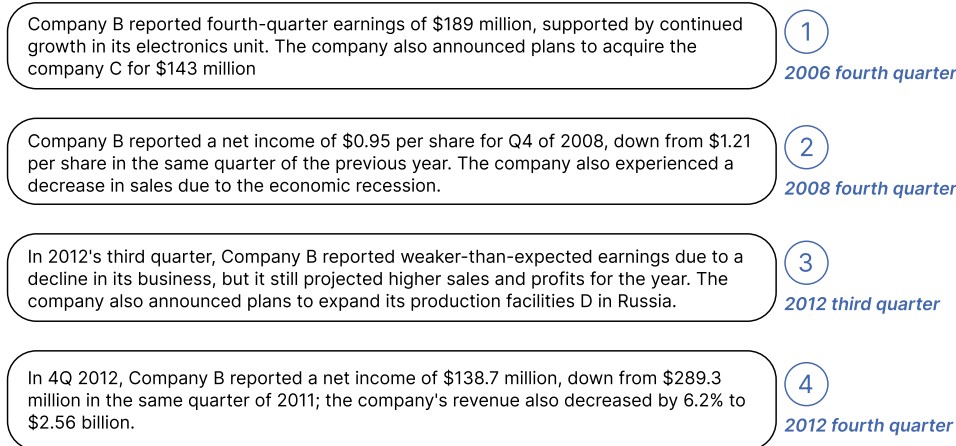

Figure 18: Example of generated contextual information for Company B marked in Figure 17

Table 12: Table of Main Notation on TEMPO

| Notation | Description |
|---|---|
| $\hat{\mathbf{x}}_t^i$ | $i^{th}$ channel prediction at time step t |
| $\mathbf{x}_t^i$ | $i^{th}$ channel look back window/historical values at time step t |
| $\Phi$ | model parameter |
| $\mathbf{V}$ | prompt value from prompt pool |
| $X$ | input data which can be decomposed into $X_T$ $X_S$ $X_R$ |
| $X_{Tt}, X_{St}, X_{Rt}$ | trend, season, residual component set in time $t$ |
| $x_{Tt}^i$ | $i^{th}$ channel $t^{th}$ timestep of $x_T^i$ |
| $\hat{\mathbf{x}}_{Tt}^i$ | predict value of trend component |
| $\mathcal{P}$ | patch of input data |
| $k_m$ | $m^{th}$ key in prompt pool |
| $V_m$ | $m^{th}$ value in prompt pool |
| $\mathbf{V_k}$ | prompt pool |
| $\mathcal{K}$ | hyperparameter, number of prompts to choose |
| $M$ | hyperparameter, length of prompt pool |
| $Z^*$ | GPT output for * (trend, seasonal, residual) |
| $L_H$ | prediction length |
| $L_E$ | embedding vector length |
| $Y_*$ | final predict value before de-normalization |
| $\hat{Y}_*$ | final predict value |

