# OpenReview forum: "TEMPO: Prompt-based Generative Pre-trained Transformer for Time Series Forecasting"
_ICLR.cc/2024/Conference — ICLR 2024 poster_

### Official Review · Reviewer_izvx · 2023-10-29

**Soundness:** 2 fair
**Presentation:** 2 fair
**Contribution:** 3 good
**Rating:** 5
**Confidence:** 4

**Summary:**

The paper proposes a way to leverage pre-trained language models for time-series forecasting. Their method is based on two key ideas: (1) decomposing time-series into trend, season and residual components can aid time-series forecasting, and (2) prompting large language models based on a shared pool of prompts.

**Strengths:**

1. Writing: The paper is well-written for the most part.
2. Interpretability: The authors aim to shed some light on the time-series predictions made by the model.
3. Modeling time-series and text together: I really liked this key insight of the paper. I think it is under-explored and valuable.
4. Theory: The authors provide some theoretical insight into the design decisions behind their model.

**Weaknesses:**

Claims:
1. While achieving strong prediction performance, the previous works on timeseries mostly benefit ... that captures temporal dependencies but overlooks a series of intricate patterns within timeseries data, such as seasonality, trend, and residual." -- I do not agree with the authors, multiple recent approaches using deep learning for time-series forecasting have decomposed model inputs into trend and seasonal components. See N-BEATS, N-HITS (stacked MLP based models), AutoFormer, as an example.
2. Theorem 3.1 -- I do not fully understand the implications of Theorem 3.1 and how that affects the design choices of the authors.
3. Prompt pool captures seasonal and trend components: The authors provide an example of 3 time-series from 1 dataset to demonstrate that the prompts capture season and trend components, but I am not sure this is sufficient evidence. It would be interesting to look at the distribution of prompts for multiple (or all) time-series in one or more datasets, as time-series are clustered based on their trend and/or seasonality components. I believe this would give a more, dataset level evidence for the authors' claims.
4. Interpretability: I am not sure how the GAM and SHAP provide interpretability, beyond confirming what is expected from these models, i.e. the residuals do not have any pattern.

Experimentation:
1. "Large-scale experiments/benchmarks": The authors omit several benchmarks, and therefore I would argue that the experiments are not large-scale. For e.g., for long horizon datasets, the authors do not use the Influenza-like Illnesses and Exchange-rate datasets which PatchTST and TimesNet, and other recent studies. Secondly, there are multiple short-horizon benchmarks, like M3 or the M4 datasets, and the much larger and comprehensive Monash Forecasting archive, yet the authors do not confirm their methods on these datasets.
2. Multiple methods are omitted from the evaluation, for e.g. statistical methods such as Auto-ARIMA, Auto-THETA, Auto-ETS etc., and deep learning methods such as N-HITS and N-BEATS. Also the authors cite PromptCast but do not compare their method to this particular baseline.
3. The value of prompt pool-- The authors demonstrate in Table 9 that the prompt pool helps model prediction. How would they explain the methods without prompt pooling doing better on some datasets for some forecasting horizons?

Clarity:
1. Insufficient details in model design and experimentation. See Questions.

Minor:
1. Typos: inclduing, outpemforms ... etc.
2. References: I would encourage the authors to find references to accepted papers, instead of citing their ArXiv versions.

**Questions:**

1. Theorem 3.1 -- I do not fully understand the implications of Theorem 3.1 and how that affects the design choices of the authors. What do we learn from the theorem?
2. Model design: The authors mention they use temporal encoding from PatchTST but do not provide any definition or references for it. As far as I am aware, PatchTST does not use temporal encoding. That is, if my understanding of temporal encoding is accurate. Moreover, work prior to PatchTST, either Informer or Autoformer use temporal encoding instead.
3. Experimentation: How are LLM-based models like BERT, T5, LLaMA used for forecasting? What is the input to these models? What is the output?
4. Experimentation: What is the context length for the long-horizon forecasting task?
5. Experimentation: How are 5/10% of the data points used? How do you define a data point in a long-horizon forecasting time-series? Also 5-10% seems like a lot of data, given that most of the time-series have multiple channels, and the model processes them independently.

---

> ### Author Response · Authors · 2023-11-20
> **Response to Reviewer izvx - "Claim"**
>
> Dear Reviewer izvx:
>
> We sincerely appreciate you taking the time to review the revisions and provide such thoughtful feedback. We have carefully considered each of your suggestions to strengthen the paper. In the updated draft, edits made specifically per your comments are highlighted in blue, while changes addressing feedback from multiple reviewers are marked in orange.
>
> [Claims W1] Accurate Claim on the Decomposition Operation:
>
> You raise a fair point that some recent works have explored decomposition techniques for time series modeling. To clarify our contribution, we provide the revised statements in the introduction:
> "While achieving strong prediction performance, some of the previous works on time series mostly benefit from the advance in sequence modeling (from RNN and GNN, to transformers) that captures temporal dependencies but have not fully capitalized on the benefits of intricate patterns within time series data, such as seasonality, trend, and residual.”
> “Methods like N-BEATS and AutoFormer decompose data but do not integrate it tightly with large pre-trained models.”
> “ In this paper, we make an attempt to go further by not only decomposing time series but also systematically incorporating the components into the transformer architecture… "
>
> [Claims W2& Q1] Design choice behind Theorem 3.1:
>
> We have expanded the discussion of Theorem 3.1 to provide more context on how it influenced our design choices in the main paper and here is a more detailed explanation:
>
> Theorem 3.1 gives the insights that if the trend and seasonal components of a time series are not orthogonal (i.e. their inner product is non-zero), then there does not exist a set of orthogonal bases that can completely disentangle and separate the two components.
> As analyzed in [1]’s Theorem1, the self-attention layer inherently learns an orthogonal transform that projects inputs into a disentangled latent space, analogous to how PCA decomposes signals into orthogonal principal components. So directly applying attention to a raw time series would fail to disentangle non-orthogonal trend and seasonal components.
> Therefore, we manually perform this decomposition using statistical methods like STL, and provide the separated components as inputs to the transformer model. This allows us to provide attention with the separated trend, seasonal, and residual components as inputs. Attention can then focus on modeling each individual component in its disentangled latent space without entanglement. Without explicit decomposition, the trend and seasonal signals would be entangled in the attention space, which will hurt the performance as shown in the Ablation study section and in Table 9.
>
>
> In summary, Theorem 3.1 formally proves the limitation of standard self-attention for time series decomposition. This motivates the design choice of using statistical decomposition techniques and providing the separated components as input to the transformer.
>
> [1] Zhou, Tian, et al. "One Fits All: Power Time Series Forecasting by Pretrained LM." Advances in Neural Information Processing Systems 36 (2023)
>
> [W3] Distribution of Prompt Pool:
>
> Thank you for raising this important point for further improve the representation of prompt pool. You are absolutely right that showing prompt selection trends across multiple full datasets would provide more robust evidence that the prompts are capturing seasonal and trend components. To clarify, the examples in Figures 8, 9, 10, 11, and 12 are actually from different datasets (ETTm1, ETTh1, ETTh2, Weather, Electricity), not a single dataset. However, we agree that a dataset-wide analysis could provide more robust evidence that this holds more broadly.
>
> In addition, we should analyze prompt selection distributions at the dataset level to support our claims. As an additional analysis, Figure 7 (new added in Appendix) plots the prompt selection distribution histograms of the selected prompts for the trend, seasonal, and residual components on ETTm1 and ETTm2 datasets. In our setting, each data can select multiple prompts (in this case, 3 prompts for each component), the frequency here is calculated by the count of each prompt selected by one data.  We can see clear differences in the prompts preferred for periodic vs seasonal components. For example, in ETTm1 prompts 11, 15, and 28 are most frequently chosen for trends while prompts 5, 7, and 18 dominate for residuals. This provides further evidence that the prompt pool can retrieve specialized prompts for distinct time series characteristics after the training stage.

---

> ### Author Response · Authors · 2023-11-20
> **Continue Response to Reviewer izvx - "Claim"**
>
> [Claim W4] The Role of GAM and SHAP:
>
> In response to your concern about the interpretability provided by Generalized Additive Models (GAM) and SHapley Additive exPlanations (SHAP), we have clarified their roles in our Appendix C in terms of beyond merely confirming the absence of patterns in the residuals. On the one hand, GAM explicitly models the different feature effects as additive components, where the interpretability is inherently incorporated into TEMPO; On the other hand, SHAP attributes feature effects after the fact to explain opaque model predictions. Thus, GAM and SHAP are powerful tools not only for confirming expected findings but also for providing deeper insights and explanations into the workings of complex models. The utility of GAM and SHAP in our analysis can be detailed as follows:
>
> First of all, the analyses quantitatively confirm assumptions about model behavior with data-driven evidence, rather than just intuition. This substantiation increases the trust and transparency in the model's predictions.
>
> Second, the component attribution could reveal unexpected behaviors if present. For example, residual impact being higher than expected could indicate overfitting noise.
>
> Third, SHAP provides nuance beyond high-level expectations, like showing the increasing error of seasonal components in longer forecasts.

---

> > ### Author Response · Authors · 2023-11-20
> > **Continue Response to Reviewer izvx - "Experimentation"**
> >
> > [Experimentation W1] "Large-scale experiments/benchmarks"
> >
> > We appreciate your insightful feedback regarding our experiment scope and the need for more comprehensive benchmarks. In our submission, we did not claim to conduct "large-scale experiments/benchmarks". However, to avoid any misunderstanding, we will substitute the term "broad" for "large-scale" in describing our evaluations. For the long-term forecasting setting, we intentionally excluded the Influenza-like Illnesses and Exchange-rate datasets due to their relatively limited time steps compared to the other seven widely used datasets. Specifically, the Exchange-Rate dataset has 8 variants with 7,588 timesteps (granularity: 1 day) and the Influenza-like Illnesses dataset has 7 variants with 966 timesteps (granularity: 1 week). These limited samples potentially constrain the activation of knowledge in a pre-trained model. Nevertheless, we emphasize the significant improvements on the datasets we did evaluate, attesting to our model's strong adaptability as shown in Section 4.3. For short-term forecasting, our goal extends beyond simply comparing results. We aim to illustrate the potential of multi-modal input in both in-domain and cross-domain sectors. Our model, TEMPO, reduces the average SMAPE error by 32.4% and 39.1% in these sectors, respectively. We focused on commonly used forecasting datasets from recent literature, but acknowledge the value of incorporating larger benchmarks, such as the M3, M4, and Monash Forecasting Archive datasets, as you've suggested. We will prioritize incorporating tests on the larger benchmark collections you mentioned in the final version. We welcome any further suggestions you may have for enhancing the experimental scales.
> >
> > [W2] Multiple Baselines
> >
> > We appreciate your insights regarding our model evaluation and the suggestion to include statistical methods like Auto-ARIMA, Auto-THETA, Auto-ETS, and deep learning methods such as N-HITS and N-BEATS.
> > It is important to note that ARIMA, N-HITS, and N-BEATS are primarily designed for univariate time series forecasting, which is not the focus of our study on multivariate long-term forecasting. The results presented in the Autoformer paper reveal that Autoformer outperforms both N-BEATS and ARIMA in univariate time series forecasting settings. Given the outperformance of our model compared to the Autoformer approach, we feel confident that our model could also outperform ARIMA, N-HITS, and N-BEATS.
> >
> > Regarding the omission of PromptCast, we acknowledge that as a prompt-based approach, it would have been a relevant comparison. However, there are key differences in our methodologies:
> > - PromptCast requires manual creation and curation of a dataset of prompts and time series for pre-training. In contrast, our method utilizes a prompt tuning strategy that automatically constructs relevant prompts by retrieving from a shared pool. This offers more flexible prompt learning, specifically tailored to the downstream forecasting task.
> > - Our method integrates decomposition with prompt tuning, which effectively incorporates numerical time series patterns into the pre-trained language model. Conversely, PromptCast applies prompts at a more superficial level without such grounded integration.
> >
> > Despite these differences, we concur that a direct comparison with PromptCast and other models would enrich our evaluation process. In future work, we plan to expand our evaluation to include classical statistical methods, established deep-learning approaches, and prompt-based competitors like PromptCast. This will provide a more comprehensive comparison and help identify our approach's unique strengths and potential limitations.
> > Thank you for bringing our attention to these additional models. We appreciate your constructive feedback and will incorporate these insights into our future research.

---

> > > ### Author Response · Authors · 2023-11-20
> > > **Continue Response to Reviewer izvx - "Experimentation"**
> > >
> > > [W3] The value of Prompt Pool
> > >
> > > In the era of large models, the notion of learning from scratch has started to give way to more efficient approaches. One of these is the utilization of prompts to guide pre-trained models, which permits focused learning from relevant knowledge, a concept derived from memory consolidation theory. Time series data are often non-stationary, meaning its properties can change over time. This presents a significant challenge for traditional models, which often struggle to adapt accordingly. The goal of the prompt pool is to direct the reusing of learnable representations at the instance level, which embodies temporal knowledge of trends and seasonality. Essentially, this allows the model to adaptively accumulate knowledge across fluctuating time distributions, thereby ensuring the model's ability to forecast is maintained as generative processes progress.
> > >
> > > While the improvement from the prompt pool in TEMPO might seem minimal at first glance, it becomes evident that the model's performance declines when the prompt pool is absent. This is particularly noticeable in the zero-shot setting, as highlighted in Tables 10 and 11 in the Appendix, where the absence of the prompt pool leads to a clear drop in TEMPO's performance. A case study provided in Appendix D.1 further illustrates the utility of the prompt pool, demonstrating its ability to retrieve similar patterns, an attribute we believe contributes to the performance enhancement. Despite these benefits, as discussed in section D.2 on 'leave-one-out' analysis, the prompt pool introduces additional parameters that require tuning. Determining optimal settings for these parameters is a complex issue that we plan to address in future research.
> > >
> > > [Minor]: We will polish it accordingly. Thanks for the suggestion!

---

> > > > ### Author Response · Authors · 2023-11-20
> > > > **Continue Response to Reviewer izvx - "Question"**
> > > >
> > > > [Q2] Model Design
> > > >
> > > > We appreciate your keen observation and the opportunity to clarify our use of the term "temporal encoding". In our work, when we refer to "temporal encoding", we mean the process of transforming temporal numerical input data into a format suitable for pre-trained language models, while preserving its temporal information. This process is inspired by the patch operation used in PatchTST.
> > > >
> > > > We recognize, however, that the term "temporal encoding" can easily be conflated with "temporal embedding", a concept used in transformer-based models such as Informer and Autoformer. The latter refers to the practice of embedding time information into the model to capture temporal dynamics, which is different from our use of the term.
> > > > To avoid any future misunderstanding, we will provide a clearer contextualization of this term in the revised version of our paper. We're grateful for your feedback, as it helps us enhance the clarity and precision of our work.
> > > >
> > > > [Q3] How are LLM-based models like BERT, T5, LLaMA used for forecasting?
> > > >
> > > > You raise an excellent question - we have provided more implementation details for the LLM baselines in our polished version in blue color. To forecast with models like BERT, we use the same overall approach as TEMPO:
> > > >
> > > > - Apply patching to convert the numeric time series into numerical vectors with temporal order and the shape of LLM’s input (Refer to the last paragraph of Section 3.2 and [1])
> > > >
> > > > - Pass the numerical vectors into the LLM to obtain a hidden representation from each LLM.
> > > >
> > > > - Project the representation through a linear layer to predict the target future values (Refer to Section 3.4).
> > > >
> > > > So the LLM takes the numerical past context as input, and outputs forecasted numeric values for the future timesteps. The key difference from TEMPO is the lack of decomposition and prompt tuning.
> > > >
> > > > [Q4] Experimentation: What is the context length for the long-horizon forecasting task?
> > > >
> > > > Thanks for raising the question about the context length details for long-horizon forecasting. We have added these details in Section D.1. Following the [1], we use a context length of 336 timesteps for all the long-term forecasting experiments.
> > > > As you noted, typical context lengths from prior work are in the range of 168-336. Informer uses 336 as one of the baseline models, so 336 is a relevant reference point.
> > > > By aligning with the context length in [1] and Informer, we aim for as fair a comparison as possible with those state-of-the-art baselines.
> > > >
> > > >
> > > > [1] Zhou, Tian, et al. "One Fits All: Power Time Series Forecasting by Pretrained LM." Advances in Neural Information Processing Systems 36 (2023).
> > > >
> > > > [Q5] Experimentation: 5%/10% data:
> > > >
> > > > We value your keen observations and agree that the terminology used in our paper requires clarification. In our research, when we refer to a "data point," we mean each individual timestep in a time series. This is in line with the standard definition used in time series analysis.
> > > > Our methodology for producing the training set employs the same technology as the One-fits-all approach. This involves taking a portion (5%/10%) of the time series data for training and using test data for testing. However, we acknowledge your concern that using 5-10% of data might seem excessive, especially considering that our model processes multiple channels independently. In our original submission, we acknowledge that the term "few-shot" might lead to potential misunderstandings in this context. As such, we have consciously avoided using "few-shot" in the main text of our paper. Instead, we have chosen "Data-efficient adaptation" as a section title, which we believe more accurately encapsulates our approach. This choice of wording more accurately reflects our approach, which aims to achieve superior performance with less data compared to traditional methods.
> > > >
> > > > In our revised paper, we will further clarify our definition of a "data point" and our methodology for determining the training set. We appreciate this opportunity to improve the clarity and accuracy of our paper.

---

> > > > > ### Comment · Reviewer_izvx · 2023-11-21
> > > > > **Response to Author Rebuttal**
> > > > >
> > > > > Dear Authors,
> > > > > Thank you for your comprehensive rebuttal. I have the following comments and questions.
> > > > > 1. Claims W1: "In this paper, we make an attempt to go further by not only decomposing time series but also *systematically incorporating the components into the transformer architecture…*" -- If Fig. 1 is a faithful representation of the methodology, then I disagree with this claim. I do not see how there are any systematic changes to the core transformer architecture. The decomposition happens in the initial layers, which is also true for other models. What am I missing?
> > > > > 2. [W1] "These limited samples potentially constrain the activation of knowledge in a pre-trained model." -- I again do not agree with this statement regarding experiments on the ILI and exchange-rate datasets. (1) I have personal experience and, (2) other papers in this very conference, using similar methods, have shown otherwise, i.e. pre-trained models can be leveraged for evaluation on these datasets. This also contradicts the few-shot experiments you have compared your methods against.
> > > > > 3. [W2] "ARIMA, N-HITS, and N-BEATS are primarily designed for univariate time series forecasting, which is not the focus of our study on multivariate long-term forecasting" -- Most if not all of the recent methods only accept univariate time-series as input, and follow the principle of channel independence to model multi-variate time-series. Moreover, AutoFormer does not compare or outperform N-HITS. Also, my suggestion for ARIMA was for short-horizon forecasting, wherein AutoARIMA and other methods have been shown to outperform deeper methods time and again.
> > > > >
> > > > > I would again like to thank the authors for their response. But I would like to stick to my current score given the questions on decomposition raised in the public comment, claims and lack of some experiments.

---

> > > > > > ### Author Response · Authors · 2023-11-22
> > > > > > **Thanks for the response!**
> > > > > >
> > > > > > Dear Reviewer izvx:
> > > > > >
> > > > > > Thanks for your further comments! We will response to them one by one:
> > > > > >
> > > > > > [1] Thank you for your inquiry. TEMPO operates as a transformer with three key distinctions from traditional transformers: (1) it decomposes the time series into trends, periods, and seasonal terms, (2) it utilizes the pre-training weights of larger language models, and (3) it incorporates a prompt pool. In this response, we'll focus on the first two distinctions, which are intended to establish an inductive bias more attuned to the characteristics of time series, as compared to transformers applied directly to raw time series input.
> > > > > > Concerning the first distinction, we illustrate through Theorem 3.1 that traditional attention may struggle with handling non-orthogonal trends and seasonal components. This observation inspired us to model the trend, seasonal, and residual elements as separate semantic entities and map them into dedicated input embeddings for the pre-trained transformer blocks.
> > > > > > In terms of utilizing pre-trained models, we recognize that the intricate patterns of timeline sequences differ from those of linguistic systems, but we also note that decomposition lessens this disparity. We look forward to delving deeper into this phenomenon in future research, both theoretically and practically.
> > > > > >
> > > > > > Given our goal to fully harness existing knowledge in the era of large models, we posit that an appropriate decomposition of time series will lay the groundwork for the construction of foundational time series models.
> > > > > > However, we want to clarify that the term "systematically" used in our text could be a miscommunication. We will update it to "systematically" in the main body of the text to better represent our meaning:
> > > > > >
> > > > > > "In this paper, we attempt to address the timely challenges of adapting large pre-trained models for time series forecasting tasks."
> > > > > >
> > > > > > We trust this response provides the clarification you were seeking.
> > > > > >
> > > > > > [2] We appreciate your perspective and wholeheartedly agree with your statement that "pre-trained models can be leveraged for evaluation on these datasets." We have indeed conducted experiments on the ILI dataset, comparing pre-trained models with the baselines.
> > > > > > In fact, when we ran the ILI dataset against pre-trained models and compared our model against both the latest state-of-the-art (SOTA) and previous work, our model performed exceptionally well, as shown in Table1. It is the new SOTA compared with large pre-trained based models and  second only to PatchTST, outperforming both Autoformer and Timesnet.
> > > > > >
> > > > > > |     | TEMPO |     | GPT4TS |     | PatchTST |     | TimesNet |     |
> > > > > > |--|-------|-----|---------|-----|----------|-----|----------|-----|
> > > > > > |     | MSE   | MAE | MSE     | MAE | MSE      | MAE | MSE      | MAE |
> > > > > > | 24  | 1.851 | 0.836 | 2.063 | 0.881 | 1.319 | 0.754 | 2.317 | 0.934 |
> > > > > > | 36  | 1.680 | 0.830 | 1.868 | 0.892 | 1.430 | 0.834 | 1.972 | 0.92 |
> > > > > > | 48  | 1.984 | 0.911 | 1.790 | 0.884 | 1.553 | 0.815 | 2.238 | 0.94 |
> > > > > > | 60  | 1.867 | 0.898 | 1.979 | 0.957 | 1.470 | 0.788 | 2.027 | 0.928 |
> > > > > > | Avg.| 1.846 | 0.869 | 1.925 | 0.903 | 1.443 | 0.797 | 2.139 | 0.931 |
> > > > > > Table 1. ILI dataset’s forecasting results.
> > > > > >
> > > > > > However, we'd like to clarify a point mentioned in our response. Different datasets possess unique feature distributions and temporal behaviors, and this diversity can potentially limit the activation of related knowledge in a pre-trained model. This statement doesn't contradict the utility of pre-trained models; rather, it underscores the potential challenges that may arise due to the specific characteristics of each dataset.
> > > > > > We welcome any additional information or insights you can provide on the state-of-the-art scheme for the ILI dataset based on a pre-trained model. This would allow us to conduct a more comprehensive and detailed analysis. Your contribution to this discussion is greatly valued.
> > > > > >
> > > > > >
> > > > > > [3] You raise fair criticism on our baseline comparisons. We agree that ARIMA, N-HITS, and N-BEATS should be included, especially given the common practice of channel independence assumptions for multivariate modeling. While Autoformer provides some initial univariate comparisons, direct benchmarking is ideal. Given the time constraints, we are currently working on long-term forecasting experiments with N-HITS and short-term results using ARIMA-based methods. We will incorporate these additional baselines into the paper as soon as the results are available. Thank you again for catching this limitation - your feedback will significantly strengthen our baseline comparisons and ensure we properly acknowledge prior state-of-the-art approaches. Please feel free to provide any other suggestions on improving the contextualization of our method's performance against relevant existing techniques.

---

> > > > > > > ### Author Response · Authors · 2023-11-22
> > > > > > > **Thanks for the response!**
> > > > > > >
> > > > > > > [4 public comments]  We have responded to the public comments, which may provide initial insight into your query. However, we are happy to go into more depth on any aspects you would like elaborated further. Please let us know how we can best address this part in a way that satisfies your interests. We are eager to have an informative discussion to ensure our approach is conveyed clearly.
> > > > > > >
> > > > > > > We sincerely thank you for the thorough review and insightful feedback on our manuscript.
> > > > > > >
> > > > > > > Best,
> > > > > > >
> > > > > > > Authors

---

> ### Author Response · Authors · 2023-11-23
> **Further results**
>
> We appreciate your insights and have thoroughly reviewed the N-HITS paper [1]. In response to your comments, we plan to include N-HITS as a robust baseline in our final paper. We will also add the following details: N-HITS introduces an innovative approach to long-horizon forecasting, addressing both computational complexity and accuracy concerns. The model's hierarchical interpolation and multi-rate data sampling techniques enable it to manage long-range dependencies, while maintaining computational efficiency, making it a potent solution for long-term forecasting.
> We have included the results of N-HiTS, N-BEATS, and ARIMA on long-term forecasting below: N-HiTS surpasses traditional transformer-based models, ranking second only to PatchTST and our model.
>
>
> |       | N-HiTS |     | N-BEATS |     | ARIMA |     |
> |-------|--------|-----|---------|-----|-------|-----|
> |       | MSE    | MAE | MSE     | MAE | MSE   | MAE |
> | ETTm2 |        |     |         |     |       |     |
> | 24    | 0.176  | 0.255 | 0.184  | 0.263 | 0.225 | 0.301 |
> | 36    | 0.245  | 0.305 | 0.273  | 0.337 | 0.298 | 0.345 |
> | 48    | 0.295  | 0.346 | 0.309  | 0.355 | 0.37  | 0.386 |
> | 60    | 0.401  | 0.413 | 0.411  | 0.425 | 0.478 | 0.445 |
> | Avg.  | 0.279  | 0.33  | 0.294  | 0.345 | 0.343 | 0.369 |
> | ECL   |        |     |         |     |       |     |
> | 96    | 0.147  | 0.249 | 0.145  | 0.247 | 1.22  | 0.814 |
> | 192   | 0.167  | 0.269 | 0.18   | 0.283 | 1.264 | 0.842 |
> | 336   | 0.186  | 0.29  | 0.2    | 0.308 | 1.311 | 0.866 |
> | 720   | 0.243  | 0.34  | 0.266  | 0.362 | 1.364 | 0.891 |
> | Avg.  | 0.186  | 0.287 | 0.198  | 0.3   | 1.29  | 0.853 |
> | Weather|       |     |         |     |       |     |
> | 96    | 0.158  | 0.195 | 0.167  | 0.203 | 0.217 | 0.258 |
> | 192   | 0.211  | 0.247 | 0.229  | 0.261 | 0.263 | 0.299 |
> | 336   | 0.274  | 0.3   | 0.287  | 0.304 | 0.33  | 0.347 |
> | 720   | 0.351  | 0.353 | 0.368  | 0.359 | 0.425 | 0.405 |
> | Avg.  | 0.249  | 0.274 | 0.263  | 0.282 | 0.309 | 0.327 |
> | ILI   |        |     |         |     |       |     |
> | 24    | 1.862  | 0.869 | 1.879  | 0.886 | 5.554 | 1.434 |
> | 36    | 2.071  | 0.934 | 2.21   | 1.018 | 6.94  | 1.676 |
> | 48    | 2.134  | 0.932 | 2.44   | 1.088 | 7.192 | 1.736 |
> | 60    | 2.137  | 0.968 | 2.547  | 1.057 | 6.648 | 1.656 |
> | Avg.  | 2.051  | 0.926 | 2.269  | 1.012 | 6.584 | 1.626 |
> | Traffic|       |     |         |     |       |     |
> | 96    | 0.402  | 0.282 | 0.398  | 0.282 | 1.997 | 0.924 |
> | 192   | 0.42   | 0.297 | 0.409  | 0.293 | 2.044 | 0.944 |
> | 336   | 0.448  | 0.313 | 0.449  | 0.318 | 2.096 | 0.96  |
> | 720 | 0.539| 0.353| 0.589| 0.391| 2.138| 0.971|
> | Avg.| 0.452| 0.311| 0.461| 0.321| 2.069| 0.950|
> Table 1. Long-term forecasting results on more baselines
>
> [1] Challu, C., Olivares, K. G., Oreshkin, B. N., Ramirez, F. G., Canseco, M. M., & Dubrawski, A. (2023, June). Nhits: Neural hierarchical interpolation for time series forecasting. In Proceedings of the AAAI Conference on Artificial Intelligence (Vol. 37, No. 6, pp. 6989-6997).
>
>
>
> For short-term forecasting, we applied ARIMA to the proposed TETS dataset on the first 7 sectors. The results, which were inferior to TEMPO, will be included in the final version of our paper.
>
> | Category               | Abs 0.8 | Abs 0.9 |
> |------------------------|---------|---------|
> | Basic Materials        | 38.7    | 40.7    |
> | Communication Services | 31.4    | 33.5    |
> | Energy                 | 56.9    | 59.8    |
> | Financial Services     | 29.5    | 30.9    |
> | Healthcare             | 13.1    | 13.1    |
> | Technology             | 17.5    | 18.7    |
> | Utilities              | 42      | 44.3    |
> Table 2. Short-term forecasting results (SMAPE) on ARIMA
>
> Thank you for engaging with us on this work - we appreciate you taking the time to thoroughly review our approach. We hope our latest responses and results help address your questions and provide satisfactory explanations about our methodology. Please don't hesitate to let us know if you have any additional queries or need clarification on particular aspects of our techniques.
>
> Best,
>
> Authors

---

### Official Review · Reviewer_W4kZ · 2023-10-31

**Soundness:** 3 good
**Presentation:** 3 good
**Contribution:** 3 good
**Rating:** 6
**Confidence:** 3

**Summary:**

Authors propose TEMPO leveraging a pre-trained language model for time-series forecasting tasks. The two main components of the proposed approach: the decomposition of time series into trend, seasonality, and residuals, as well as the prompt learning, effectively increase the forecasting performance. The improvement is significant. Also, the paper demonstrates the ability of the proposed method to be trained with few-shots and to be adapted to unseen datasets.

**Strengths:**

1) The paper is well-written.
2) The improvement of forecasting performance over benchmark methods is significant and consistent across datasets.

**Weaknesses:**

1) The idea of decomposition of trend, seasonality, and residuals is not that novel and has been used for time-series forecasting.
2) The theorem 3.1 does not directly prove the point “more importance in current transformer-based methods as the attention mechanism, in theory, may not disentangle the disorthogonal trend and season signals automatically".
3) The result shown in Figure 2 seems to be obvious since the trend is easier to learn and may take a large portion of the data.

**Questions:**

1) In the few-shot learning setting, do you need to finetune the model on other time-series datasets first?
2) What is the computation complexity of the proposed method, and how it is comparable to other methods?
3) Based on the ablation study in Table 9, it seems that w/o prompts have less effect on the final performance. Sometimes without prompts, the performance can be even better. Could you explain this?
4) How the prompts are initialized and trained.
5) In Table 1, why don’t you include the traffic forecasting results?
6) How generalizable can the model be for the unseen domain transfer? For example, what if you choose weather and traffic as the training domain then apply the model to EETm1 and EETm2

---

> ### Author Response · Authors · 2023-11-20
> **Response to Reviewer W4kZ**
>
> Dear reviewer W4kZ,
>
> We sincerely appreciate your kind recognition of our work and your positive feedback on the improvements made to our paper. We have carefully considered each of the points you raised and would like to respond to them individually. To further address your request, we have made specific changes to the revised version of the paper. We have highlighted the sections that directly correspond to your suggestions in green, emphasizing the implementation of your specific recommendations. Additionally, we have highlighted sections that address common suggestions in orange, indicating the incorporation of general feedback.
>
> [W1] Novelty for the decomposition:
>
> While decomposing time series into trend, seasonality, and residuals has been explored before, the innovation in this work lies in the unique integration into a pre-trained transformer model paired with a prompt tuning strategy. Theoretical analysis reveals standard attention may fail to disentangle non-orthogonal trend and seasonal components, common in real data. By leveraging classical STL decomposition and mapping components into separate hidden spaces, the model capitalizes on the semantic reasoning of pre-trained models like GPT to understand each individually.
>
> This principled input representation allows the transformer to capture intricate time series dynamics across diverse applications. The prompt pool further enhances adaptability, enabling instance-level tuning to consolidate shifting temporal knowledge. The model can recall relevant learned representations encoded in prompts based on input similarity.
> In essence, while decomposition and prompting have been separately studied, their novel combination here to unlock the capabilities of pre-trained transformers is a key innovation. The approach is grounded in both theory and empirical results. While the individual pieces may not be wholly novel in isolation, their synergistic integration to advance time series modeling with modern techniques represents a significant contribution. The improvements in accuracy and data efficiency validate the benefits of unifying classical decomposition with prompt-based transfer learning in pre-trained transformers.
>
> [W2] To clarify the theorem 3.1:
>
> We clarified the theorem 3.1 in the main paper in orange color.
>
> Theorem 3.1 establishes that when the trend T(t) and the seasonality S(t) are not orthogonal, there cannot exist a set of orthogonal bases that can disentangle S(t) and T(t) into two disjoint sets of bases. It's important to note that it's quite common that the trend and seasonal components are not independent of each other in many real-world time series, and this interdependence needs to be accounted for when decomposing the time series for analysis and forecasting.
>
> The reason for this lies in the nature of their spectrums: the spectrum of a periodic signal is discrete, while the spectrum of a non-periodic signal is continuous. Consequently, overlaps on the non-zero frequencies of the periodic signal are quite likely.
> Principal Component Analysis (PCA), which also aims to learn a set of orthogonal bases on the data, thus, cannot disentangle these two signals into two separate sets of bases. Drawing from [1] Theorem 1, it becomes evident that the self-attention mechanism in pre-trained large models executes a function bearing a strong resemblance to PCA. Hence, unless this operation is manually carried out, the self-attention mechanism cannot automatically decompose the time series into its trend and seasonal components when there exists inherent non-orthogonality of these components.
>
> Note that, this theoretical understanding can provide insights on why manual decomposition can boost the model's performance. To make better use of the semantic capabilities of the pre-trained model, we propose conducting a decomposition operation on the time series. This operation aims to separate the orthogonal/non-orthogonal trend and seasonal components, thereby enhancing the model's ability to understand and predict the time series data.
>
> [1] Zhou, Tian, et al. "One Fits All: Power Time Series Forecasting by Pretrained LM." Advances in Neural Information Processing Systems 36 (2023)

---

> > ### Author Response · Authors · 2023-11-20
> > **Continue Response to Reviewer W4kZ**
> >
> > [W3] SHAP value
> >
> > We have addressed the role of SHapley Additive exPlanations (SHAP) value in Appendix C in our revised version. Here, we briefly summarize the answer as follows:
> >
> > SHAP values provide a comprehensive quantification of the importance of each feature. They measure the average contribution of each feature to the prediction output across all potential feature combinations. In time series analysis, it's intuitive to assume that the trend significantly influences the prediction process. However, we need a more nuanced understanding of how various types of component errors increase, leading to a decrease in the final prediction accuracy, particularly as the prediction length expands.
> >
> > For instance, in datasets exhibiting strong seasonality, the seasonal component may display much larger variations than the residual component. Conversely, in datasets with minimal seasonality, the variations between these two components should be more comparable. We can calculate the strength of seasonality via: $\mathrm{S}=\max \left(0,1-\frac{\operatorname{Var}\left(R_t\right)}{\operatorname{Var}\left(S_t\right)+\operatorname{Var}\left(R_t\right)}\right)$. When we compare the seasonality strengths of different datasets, we find that ETTm1 (as shown in Figure 2, with a seasonality strength of 0.99) constitutes strongly seasonal data, whereas the weather dataset (depicted in Figure 13 with a seasonality strength of 0.476) exhibits less seasonality and a more pronounced trend.
> > These findings align with the conclusions drawn from the SHAP values. The performance degradation of ETTm1, when the prediction length is increased, can be primarily attributed to inaccuracies in the prediction of seasonal terms. In summary, SHAP provides insights into how the model arrives at the predictions and how it can be further improved. The ability to discern how much and where components contribute enables targeted improvements. These insights can guide us in better leveraging inductive bias to enhance both efficiency and effectiveness in the era of pre-training models. One of the interesting future works is that we can adaptively and selectively optimize specific components based on the SHAP scores during the training process. This approach would allow us to focus our computational resources and efforts on the most influential components, thereby improving the overall effectiveness of the model.
> >
> > [Q1] Experiment setting:
> >
> > You raise an excellent question about the experimental setup. To clarify, in the data-efficient adaptation experiments in Section 4.2, we do not perform any fine-tuning of the model on other time series datasets. For each experiment, only a small fraction of the target dataset's training data (e.g. 5%, 10%) is utilized, without additional data from other domains.
> > In addition, we evaluate our model's zero-shot learning capabilities in two different settings without any fine-tuning:
> > - In Section 4.3, the model is trained on 5 time series datasets, then directly tested on ETTm2 and Traffic without ever seeing those datasets.
> > - In Section 4.4, the model trained on in-domain data is zero-shot tested on the unseen cross-domain sectors of the proposed TETS dataset.
> >
> > In both cases, the model is not fine-tuned or adapted using any examples from the test datasets. The strong performance demonstrated highlights the model's ability to generalize and transfer knowledge to entirely new distributions without dataset-specific fine-tuning.
> >
> > [Q2]  Computation Complexity:
> > The computational complexity of the proposed TEMPO model can be analyzed as follows:
> > - Decomposition into trend, seasonal, residual via STL has complexity O(n), where n is the length of the input time series.
> > - Mapping components to hidden spaces is just linear transformation, also O(n).
> > - Prompt pool selection is O(nm), where m is the number of prompts, based on similarity scoring.
> > - The self-attention layers in the GPT transformer backbone dominate complexity. Each layer is O(n^2 * d) for sequence of length n and hidden dimension d. With L layers, the overall complexity is O(Ln^2*d).
> >
> > So the total complexity is O(Ln^2*d) as the transformer layers dominate. This is comparable to state-of-the-art transformers like Informer, and PatchTST, which also have quadratic self-attention complexity. As shown in Figure 15, TEMPO also has lower training time than models like LLAMA, TimeNET and FEDformer. So, while TEMPO has a comparable order of complexity to related transformers, it provides accuracy, capability and efficiency benefits from the unique proposed components. The overhead of decomposition and prompting is minor compared to the self-attention layers.

---

> > > ### Author Response · Authors · 2023-11-20
> > > **Continue Response to Reviewer W4kZ**
> > >
> > > [Q3] The improvement of Prompt Pool:
> > >
> > > As also mentioned by Reviewer 2NZn, the marginal gains from adding the prompt pool in some cases, seem small given its intended benefits. We want to address that dataset characteristics may favor the specialized modeling of other methods over TEMPO's more generalizable approach in some cases. For example, highly linear or simple patterns may benefit from simpler baselines. Here are a few potential reasons the prompt pool improvements may be weaker than expected:
> > >
> > > - The pre-trained model itself already encodes substantial temporal knowledge, limiting the additional value prompts can provide.
> > > - The decomposition and input representation techniques may already equip the model to handle trends, seasonality, etc well - reducing reliance on prompts.
> > > - The decomposition may not be optimal for all time series if there are more complex nonstationary patterns. Our current prompt design focuses on trends and seasonalities.
> > > - The prompt tuning method may not be optimizing and selecting prompts from the pool as effectively as possible. Suboptimal prompts could limit gains.
> > > - Prompts help most for complex cases like sparse data or domain shifts. Many evaluation datasets may be too dense/similar for prompts to shine.
> > > - The power of prompts manifests more in their scalability and generalization ability rather than raw performance gains. Their value may be more in transferability.
> > >
> > > In summary, the intuitive benefits of the prompt pool may be dampened by the model's existing capability, dataset characteristics, prompt optimization difficulties, and other factors. But your observation highlights opportunities to improve prompt integration to unlock their full potential.
> > >
> > > [Q4] Initialization and Training of Prompt Pool:
> > >
> > > We added more information on the training and initialization of the prompt pool in Appendix E.4 in green color according to your suggestion. Here is a summary of how the prompts are initialized, trained, and designed in our work:
> > > - Initialization: The prompt embeddings in the pool are randomly initialized from a normal distribution, as is standard practice for trainable parameters in neural networks.
> > > - Training: The prompts are trained in an end-to-end manner along with all other model parameters to optimize the forecasting objective. This allows the prompts to be continuously updated to encode relevant temporal knowledge.
> > > - Design: The time series retrieval-based prompt pool length is selected via grid search based on performance. We analyze different design choices for prompt type on TETS dataset: prompt pool, hard prompt and soft prompt, etc. in Appendix D.3 (Table 12, 13).
> > > - Analysis: Appendix D.4 provides analysis on prompt effectiveness by visualizing prompts selections and similarity. We find prompts successfully retrieve related representations for time series with similar patterns.
> > >
> > > In summary, the prompts are standard randomly initialized embeddings that are trained end-to-end. The number of prompts and embedding dimensions are treated as hyperparameters and tuned for good performance. Detailed design analysis provides insights into prompt similarity and selection. Our work offers an initial exploration into prompt-based tuning for time series forecasting, but substantial room remains for advancing prompt pool design. The number and structure of prompts can be investigated, along with initialization schemes, similarity scoring functions, and prompt training strategies. Establishing more rigorous design principles for prompt pools tailored to temporal data would be a valuable direction for future investigation, and could uncover techniques for maximizing the effectiveness of prompt-based knowledge transfer.
> > >
> > > [Q5] Traffic dataset:
> > >
> > > Due to constraints on space in the current version of the paper, we have not included the traffic forecasting results in Table 1. However, we appreciate the importance of these results for a comprehensive understanding of the method's performance. Therefore, we intend to incorporate them (in Table 5 in Appendix) into the final version of the paper. This addition will ensure a more thorough evaluation and discussion of our proposed method's performance across various applications.

---

> > > > ### Author Response · Authors · 2023-11-20
> > > > **Continue Response to Reviewer W4kZ**
> > > >
> > > > [Q6] Unseen Domain Transfer
> > > >
> > > > The strong performance shown in Sections 4.3 and 4.4 illustrates the model's capability for generalizing to entirely new and unseen domains without any fine-tuning. For example, in Section 4.3 the model trained on datasets like ETT and Weather generalizes well to diverse unseen datasets like ETTm2 and Traffic. While we did not explicitly test weather/traffic to ETT, based on the zero-shot transfer results, we can reasonably expect competitive performance in this setting as well. In our work, the decomposition and prompting techniques equip the model to extract meaningful representations even for out-of-domain data.
> > > > More rigorously evaluating the model's zero-shot learning capacity across diverse domain pairs is an interesting future direction. However, the current results provide promising indications that the model can generalize broadly to unseen distributions. The prompt tuning in particular allows flexibly retrieving relevant knowledge from past experiences.
> > > >
> > > > However, it is worth noting that this transfer is likely to be more effective if there are shared patterns across the datasets. If the training domains (e.g., weather and traffic) share underlying temporal or structural similarities with the target domain (e.g., ETTm1 and ETTm2), the model is better positioned to apply learned insights to the new context. In other words, the strength of the transfer can be influenced by the degree of commonality in patterns across the different domains. Further, adjustments and fine-tuning may be necessary to optimize performance in some cases. Despite these potential challenges, our findings in Section 3.3 and Section 3.4 provide promising evidence of the model's adaptability and robustness in domain transfer scenarios.

---

### Official Review · Reviewer_2NZn · 2023-11-01

**Soundness:** 4 excellent
**Presentation:** 3 good
**Contribution:** 4 excellent
**Rating:** 8
**Confidence:** 4

**Summary:**

This paper proposes a new interpretable solution, TEMPO, for time series forecasting using the power of pre-trained generative models. Specifically, this work first addresses that the semantic information inside time series is important, for example, the authors utilize the trend, seasonality, and residual information to build the tokens of the pre-trained generative models. In addition, this paper proposes a prompt pool to memory the historical patterns of different time series. The reasonable design and results indicate that TEMPO paves the path of further exploring the pre-training models’ power for time series problems.

**Strengths:**

1. This paper’s writing is clear and easy to follow. For example, the methodology part gives a clear description on how to build the time series input representation and the design of the prompt pool. The experiments on long-term time series forecasting, short-term forecasting and towards foundation model’s training are well organized to prove the model’s power from different aspects.
2. The proposed solution is well motivated: the motivation of decomposition is supported by both empirically and theoretically and the introduce of retrieval-based prompt selection can help the large pre-trained model handle complex non-stationary time series data with distribution shifts.
3. Utilizing the pre-trained transformer backbone, TEMPO give a state-of-the-art results on the popular time series research dataset.

**Weaknesses:**

1. The prompt pool’s improvement is limited: the prompt pool is supposed to have more contribution to the accuracy as the intuition is clear and convincing.
2. The collection of TETS dataset is not clear: a clear but simple discription in the main paper is necessary.
3. It seems only decoder-based pretrain model is considered in this paper. The encoder based backbone (like Bert) and encoder-decoder based backbone (like T5) is also recommend in this stage.

**Questions:**

1. Do you have any insights on the prompt pool’s marginal improvement, which is somehow against the intuition;
2. The description of TETS dataset is suggested to include into  the main paper;
3. Can you provide the reason on why only decoder-based GPT is included in the experiments?

---

> ### Author Response · Authors · 2023-11-20
> **Response to Reviewer 2NZn**
>
> Dear Reviewer 2NZn,
>
> Thank you for providing us with constructive comments and insightful suggestions to enhance the presentation of our paper. We appreciate your valuable input, and we have carefully reviewed your feedback. In response, we would like to address each point raised:
>
> 1. [W1 & Q1] Insights on the prompt pool:
>
> You raise an excellent point - the marginal gains from adding the prompt pool in some cases, seem small given its intended benefits. We want to address that dataset characteristics may favor the specialized modeling of other methods over TEMPO's more generalizable approach in some cases. For example, highly linear or simple patterns may benefit from simpler baselines.
> Here are a few potential reasons the prompt pool improvements may be weaker than expected:
> - The pre-trained model itself already encodes substantial temporal knowledge, limiting the additional value prompts can provide.
> - The decomposition and input representation techniques may already equip the model to handle trends, seasonality, etc well - reducing reliance on prompts.
> - The decomposition may not be optimal for all time series if there are more complex nonstationary patterns. Our current prompt design focuses on trends and seasonalities.
> - The prompt tuning method may not be optimizing and selecting prompts from the pool as effectively as possible. Suboptimal prompts could limit gains.
> - Prompts help most for complex cases like sparse data or domain shifts. Many evaluation datasets may be too dense/similar for prompts to shine.
> - The power of prompts manifests more in their scalability and generalization ability rather than raw performance gains. Their value may be more in transferability.
>
>
> 2. [W2 & Q2] Adding TETS dataset's description:
>
> Thank you for the suggestions on improving the presentation of our paper. We added the summary of our proposed TETS dataset in Appendix H  (in red color) and will add it in the main paper once we have extra space:
>
> The TETS dataset is a financial dataset primarily sourced from the quarterly financial statements of the 500 largest U.S. companies that make up the S\&P 500 index. It spans 11 sectors, including Basic Materials, Communication Services, Energy, Financial Services, Healthcare, Technology, Utilities, Consumer Cyclical, Consumer Defensive, Industrials, and Real Estate. To create a diverse task setting, the dataset is divided into training and evaluation sectors (the first seven sectors), and unseen sectors (the last four) for zero-shot forecasting. Sliding Window[1] is used to build the time series samples, where we can get 19,513 samples from 7 sectors as seen data for training and testing and the other unseen data are only used for zero-shot testing.
>
> The process of collecting and summarizing this contextual information is facilitated by the ChatGPT API from OpenAI. It effectively generates the desired contextual information based on consolidated details such as company information, query, quarter, and yearly context. Neutral contextual information is denoted by 'None' when the API fails to generate any context. The TETS dataset thus offers a comprehensive view of both time series data and their corresponding contextual texts for each company.
>
>
> [1] Antony Papadimitriou, Urjitkumar Patel, Lisa Kim, Grace Bang, Azadeh Nematzadeh, and Xiaomo Liu. 2020. A multi-faceted approach to large-scale financial forecasting. In Proceedings of the First ACM International Conference on AI in Finance. 1–8.
>
> [W3 & Q3]: More results.
>
> Thank you for raising this important point. You are absolutely right that we should have included results for encoder-only and encoder-decoder models in addition to the decoder-only GPT architecture. Here are a few key reasons we focused solely on GPT initially:
>
> - GPT has shown strong performance on auto-regressive forecasting tasks in prior work, so we wanted to validate its efficacy with our approach.
>
> - The decoder-only design fits naturally with the auto-regressive forecasting formulation.
>
> However, please refer to the Tables 5, 6, and 7 in the appendix for the encoder-based (Bert) model’s results and the encoder-decoder-based (T5) model’s results. T5 and Bert perform better than GPT4TS on the Weather, Traffic and ECL datasets, but worse on ETT* dataset.

---

### Public Comment · ~Zezhi_Shao1 · 2023-11-17
**The effectiveness of decomposition.**

Thank you for your excellent work!

TEMPO has made great progress in many datasets, but one thing that puzzles me is the effectiveness of decomposition.
In Table 9, the ablation experimental studies show that the decomposition mechanism, which has been widely adopted in previous work (e.g. DLinear), plays the most important role in TEMPO. Without decomposition, the performance of TEMPO appears to be close to that of existing methods, e.g. DLinear, PatchTST. In contrast, the prompt mechanism, which appears to be one of the most important technical contributions, has a limited contribution.
These results are very confusing to me, am I missing something or misunderstanding something?
Thank you again for your excellent work. I can't wait to try TEMPO in more datasets and scenarios.

Best,

Zezhi Shao

---

> ### Author Response · Authors · 2023-11-22
> **Thanks for your interest!**
>
> Hi Zeshi,
>
> Thank you for your insightful question! At its core, TEMPO operates as a transformer with three critical distinctions from traditional transformers: (1) it decomposes the time series into trends, periods, and seasonal terms, (2) it employs the pre-training weights of larger language models, and (3) it integrates a prompt pool.
>
> With regard to the first distinction, we elucidate this using Theorem 3.1. This theorem posits that when the trend T(t) and the seasonality S(t) are not orthogonal, it's impossible to find an orthogonal set of bases that can separate S(t) and T(t) into two disjoint sets of bases. It's crucial to note that the trend and seasonal components often intermingle in many real-world time series. This interdependence must be accounted for when decomposing the time series for forecasting and analysis. As suggested by [1] Theorem 1, the self-attention mechanism in pre-trained large models performs a task akin to PCA, striving to learn a set of orthogonal bases on the data. Hence, it cannot separate these two signals into distinct sets of bases.
>
> Regarding the use of pre-trained models, we surprisingly found their impact greatly surpasses the baseline. We attribute this primarily to the inability of attention to automatically decompose these patterns. Additionally, we believe that the intricate patterns of the timeline sequence differ from those of linguistic systems, with decomposition reducing this difference. For instance, due to their repetitive patterns, large models may perceive periodic items as rhythmic data. We plan to explore this phenomenon more in future research, both theoretically and practically.
>
> Moreover, the prompt pool enhances adaptability, allowing for instance-level tuning to integrate evolving temporal knowledge. Based on input similarity, the model can recall relevant learned representations encoded in prompts. We wish to point out that the characteristics of specific datasets might occasionally favor the specialized modeling of other methods over TEMPO's more comprehensive approach. For instance, highly linear or simplistic patterns may benefit more from simpler baselines. Here are a few potential reasons why the improvements from the prompt pool might not be as substantial as expected:
>
> - The pre-trained model may already encode a significant amount of temporal knowledge, thereby diminishing the additional value that prompts can provide.
> - The decomposition and input representation techniques might already equip the model to handle trends, seasonality, etc., effectively, reducing the need for prompts.
> - The decomposition might not be ideal for all time series, particularly those with more complex nonstationary patterns. Our current prompt design primarily focuses on trends and seasonalities.
> - The prompt tuning method may not be optimizing and selecting prompts from the pool as effectively as possible. Suboptimal prompts could limit gains.
> - Prompts are most beneficial for complex cases such as sparse data or domain shifts. Many evaluation datasets may be too dense or too similar for prompts to be effective.
> - The power of prompts is more evident in their scalability and generalization ability rather than raw performance gains. Their value may lie more in transferability.
>
> Considering the foundational model of time series, we believe the decomposition of time series and the use of retrieval-based prompt pools to guide models using appropriate knowledge will become the paradigm for future large time series models. Thanks again for your comments to make our paper more valuable and clearer!
>
> [1] Zhou, Tian, et al. "One Fits All: Power Time Series Forecasting by Pretrained LM." Advances in Neural Information Processing Systems 36 (2023)
>
> Best,
>
> Authors

---

### Public Comment · ~Arindam_Jati1 · 2023-11-21
**Curious about decomposition method**

It's a nice paper with good results. However, I too have the same confusion on the improvements w.r.t. the decomposition as already mentioned by others in the public comment.

From the ablation study, the improvement proposed in this paper is primarily because of decomposition. However, decomposition is tried by several others in past. `DLinear` tried trend and residual decomposition but didn't have seasonal part in it. Hence, we tried replicating the approach proposed in this paper using the STL decomposition from `statsmodel` and learnable prompt infusion from L2P paper/code(CVPR 22). However, the numbers do not improve as compared to existing SOTA as mentioned in the paper. However, while doing the decomposition, if we do two-sided decomposition or decomposition on the entire data before `X,y` windowing, then the results improve. However, this way of decomposition is leaking the future data which is not correct. Hence the confusion. Are we sure that, the decomposition do not use any data from test-set? STL decomposition from `statsmodel` have only `fit` and no `transform`. So, how are we doing the STL fit on train data and only transform on the test data? Requesting authors to clarify the approach followed for decomposition and also share code if possible. We are eager to try this method on many datasets and applications. Thank you.

---

> ### Author Response · Authors · 2023-11-22
> **Thanks for your question!**
>
> Dear Arindam,
>
> Thank you for your interest in our work and for raising this important question about the decomposition methodology. We appreciate you taking the time to try replicating our approach - your feedback will help us improve the clarity of our methodology. You can easily implement the STL algorithm according to [1], where they provide two ways to realize the STL based on the principle of fast computation of loess. The parameters of the algorithm can be obtained by cross-validation. In practice, one can choose the weighted linear regression to mimic loess so that we can have the transform ability. We are working on cleaning up and documenting the code to make our implementation more clear. We plan to open-source it with details once it is ready after the double-blind stage. Thank you again for your interest and for catching this critical detail!
>
> [1] RB, CLEVELAND. "STL: A seasonal-trend decomposition procedure based on loess." J Off Stat 6 (1990): 3-73.
>
> Best,
>
> Authors

---

> ### Public Comment · ~Lifeng_Shen1 · 2023-11-23
> **An intriguing observation to discuss**
>
> This is an interesting work, but I have the same confusion about the improvements in the results. The performance improvement on some datasets is up to *two orders of magnitude* compared to SOTA methods, especially on the complex *Weather* dataset, which is surprising.
>
> The experiment by Arindam on *decomposing the entire data before X, y windowing* provides an intriguing observation, despite incorrectly leaking the future data. Here, I would like to share another interesting experiment on future data leaking when windowing. When performing instance normalization on X using both X and Y together or normalizing X using the mean/std of Y, there is also a significant improvement in performance by an order of magnitude. However, I haven't fully understood why this has such a big impact. Do you have any insights on this issue?

---

> ### Public Comment · ~Mikhail_Seleznyov1 · 2024-04-04
> **Joining to questions about decomposition**
>
> I want to second Arindam Jati, Lifeng Shen and Zezhi Shao on the question about the surprising effectiveness of the decomposition method. Last time I've seen a similarly large jump in performance improvements in long-term time series forecasting on this benchmark was in [1], and the impact of those results on the field was very profound. This paper has a potential to have the same kind of impact, so we all would be grateful to get more details on how the decomposition is used, and to look into the implementation.
>
> [1] Are Transformers Effective for Time Series Forecasting?, https://arxiv.org/abs/2205.13504v3

---

### Meta-Review · Area_Chair_Apfe · 2023-12-08

**Metareview:**

This paper proposes an approach that combines pre-trained language models with time series modeling to make time series predictions.The idea of using language models in forecasting is quite interesting and the proposed approach shows consistent performance improvement and effectiveness in few-shot learning.

**Justification For Why Not Higher Score:**

The idea of the proposed method itself is interesting and worth sharing, but not so much as a significant technical contribution.

**Justification For Why Not Lower Score:**

Interesting idea and deserves to be shared in the community.

---

### Decision · Program_Chairs · 2024-01-16

Accept (poster)